# Patient-derived lung cancer organoids as in vitro cancer models for therapeutic screening

Minsuh Kim[1], Hyemin Mun[1], Chang Oak Sung[1,2], Eun Jeong Cho [1], Hye-Joon Jeon[1], Sung-Min Chun[1,2], Da Jung Jung [3], Tae Hoon Shin [3], Gi Seok Jeong[3], Dong Kwan Kim[4], Eun Kyung Choi[5], Seong-Yun Jeong[5], Alison M. Taylor[6], Sejal Jain[6], Matthew Meyerson [6] & Se Jin Jang[1,2]

Lung cancer shows substantial genetic and phenotypic heterogeneity across individuals, driving a need for personalised medicine. Here, we report lung cancer organoids and normal bronchial organoids established from patient tissues comprising five histological subtypes of lung cancer and non-neoplastic bronchial mucosa as in vitro models representing individual patient. The lung cancer organoids recapitulate the tissue architecture of the primary lung tumours and maintain the genomic alterations of the original tumours during long-term expansion in vitro. The normal bronchial organoids maintain cellular components of normal bronchial mucosa. Lung cancer organoids respond to drugs based on their genomic alterations: a BRCA2-mutant organoid to olaparib, an EGFR-mutant organoid to erlotinib, and an EGFR-mutant/MET-amplified organoid to crizotinib. Considering the short length of time from organoid establishment to drug testing, our newly developed model may prove useful for predicting patient-specific drug responses through in vitro patient-specific drug trials.

[1] Asan Center for Cancer Genome Discovery, Asan Institute for Life Sciences, Seoul, South Korea. [2] Department of Pathology, Asan Medical Center, University of Ulsan College of Medicine, Seoul, South Korea. [3] Biomedical Engineering Research Center, Asan Institute for Life Sciences, Seoul, South Korea. [4] Department of Thoracic Surgery, Asan Medical Center, University of Ulsan College of Medicine, Seoul, South Korea. [5] Department of Radiation Oncology, Asan Medical Center, University of Ulsan College of Medicine, Seoul, South Korea. [6] Department of Medical Oncology and Center for Cancer-Genome Discovery, Dana-Farber Cancer Institute and Department of Pathology, Harvard Medical School, Boston, MA, USA. Correspondence and requests for materials should be addressed to S.J.J. (email: jangsejin@amc.seoul.kr)

Lung cancer is the leading cause of cancer mortality worldwide[1]. Lung cancer is histologically diverse and includes three major types (adenocarcinoma, squamous cell carcinoma, and small cell carcinoma) and several less frequent types (including adenosquamous carcinoma and large cell neuroendocrine carcinoma). Standard molecular-targeted therapies have been developed according to genetic alterations, such as *EGFR* mutations and *ALK* fusions[2], and expression of biomarkers, such as programmed death-ligand 1 (PD-L1)[3]. Classical in vitro models, including cancer cell lines, have been essential to investigate molecular-targeted therapy based on genetic alterations, as they provided considerable advantages in terms of manipulation, time, and throughput[4]. However, large-scale genomic analyses of lung cancer demonstrate phenotypic and genomic diversity between individual patients represented by intertumoural and intratumoural heterogeneity[5]. Cancer cell lines do not generally maintain their original heterogeneity and three-dimensional (3D) organ structure; therefore, they are fundamentally limited in representing the complexity of lung cancer. Patient-derived xenograft models (PDXs) recapitulate the original cancer in terms of tissue structure[4,6,7] and maintain the genetic and histological characteristics of the original cancer for up to 14 passages[8]. However, PDXs have several disadvantages: establishing a PDX model has a low success rate (average 30–40%)[7] and requires a long time (from 2 to 10 months)[6,7,9]. In addition, a PDX model is costly and resource-intensive, limiting statistical power, as well as the potential for high throughput studies[10].

Recently, tissue-specific stem cells derived from several adult human organs, such as colon[11], stomach[12], liver[13], pancreas[13], and lung[14], have been cultured in 3D conditions using hydrogel with collagen or other ECM components, such as Matrigel. In these conditions, cells proliferate and give rise to differentiated progeny that undergo self-organisation[15]. This characteristic arises from specific organoid culture conditions, which promote stem cell proliferation and differentiation[16]. Therefore, organoids represent a functional unit that consists of a hierarchy of stem cells and differentiated cells[16,17]. Human cancer also proliferates and organises into specific tissue architecture in vivo, so cancer cells derived from several human primary tumour samples, including colon[18], pancreas[19], prostate[20], liver[21], breast[22], and bladder[23] cancers, have been used to establish organoids that successfully recapitulate the cancer tissue architecture. Moreover, cancer organoids require less time to become established than PDXs and have been shown to stably maintain morphological and genetic features even after long-term expansion[11,22]. Therefore, cancer organoids derived from human cancer tissues have been suggested as alternative in vitro models that maintain the characteristics of the original tumours and can potentially serve as avatars for selecting anticancer therapeutics[24] and biobanking for individual patients.

Organoid models for both tracheobronchial and alveolar tissue have been developed from adult stem cells or pluripotent stem cells[25–27] and shown to recapitulate the epithelial organisation of these two distinct tissue types in vitro[14,26,28]. As personalised models for lung cancer, in vitro tissue culture[29] or tumour spheroid culture[10,30] using a 3D culture system has been studied to predict response to anticancer therapy, but have limited growth or recapitulation of original tumour architecture[23]. More recently, lung cancer organoids (LCOs) culture using the airway organoid culture system has been reported. But there were some limitation to lung cancer specific growth and to represent variety of lung cancer[31]. Here, we report the derivation of organoids from primary lung cancer tissues (LC tissues) and paired non-neoplastic airway tissues, creating a biobank of 80 LCO lines from five subtypes of lung cancer and five normal bronchial organoids (NBOs). We demonstrate that our newly established biobank of LCOs and NBOs faithfully maintain the histological and genetic characteristics of their respective parental tissues and have potential for use in patient-specific drug trials and proof-of-concept studies on targeted therapy and resistance mechanisms.

## Results

**LCOs are established from five subtypes of cancer tissues.** To establish LCOs from LC tissues, we developed a 3D culture protocol in Matrigel (Corning, NY, USA) using minimum basal medium (MBM), which is a suboptimal media inhibiting growth of normal cell due to depletion of Wnt3a and Noggin. MBM was modified from the media for lung tumour initiating cells containing epidermal growth factor (EGF), basic fibroblast growth factor (bFGF) and N2 supplement for insulin and transferrin[32,33]. To establish LCO in 3D culture, surgically resected LC tissues were dissociated as individual cells or cell clusters, embedded in Matrigel, and submerged in MBM. Even though our MBM contained fewer reagents and growth factors compared to other protocols, which contain several reagents and growth factors including FGF10[26–28], FGF4[27], FGF7[28], Noggin[26–28], and Wnt3a[26–28], LCOs derived from LC tissues consistently generated round shapes (Fig. 1a). As shown in Fig. 1b, these LCOs were cultured for long-term expansion over 6 months without any change in spherical organoid morphology, and maintained proliferation capacity measured by a marker Ki67 immunolabelling (Supplementary Fig. 1a–c). In addition, we tried organoid culture with a small biopsy tissue to determine whether our culture protocol is applicable to samples obtained from a non-operable clinical setting. As shown Supplementary Fig. 1d, the cells from a biopsy successfully cultured to form organoids.

To assess the efficiency of generating LCOs in comparison with other patient-derived cancer models (2D cultured lung cancer cells and PDXs), we established these three models in parallel from 36 patient tumour tissues of five different histological subtypes as a starting cohort (Table 1). Two-thirds of each LC tissue was transplanted into an immunodeficient mouse to develop a PDX, and the remaining one-third was divided equally to culture cells in 2D and LCOs. Due to variation in tumour purity, some samples failed to form LCOs (Fig. 1c). Samples that failed to establish LCOs also failed to establish PDXs (Table 1). Thus, we developed a tissue quality evaluation protocol before culturing LCOs. After dissociating the LC tissues using collagenase, an aliquot of cell suspension was prepared on cytologic slides and stained with H&E to evaluate the proportion of viable epithelial-like cells (Supplementary Fig. 2). As a result, we grouped 23 epithelial cell predominant samples, four fibroblast predominant samples and nine acellular samples (Fig. 1d and Table 1). Epithelial cell predominant samples were used as starting material for 2D cultured primary lung cancer cells (2D-LC cells), organoid culture and PDX. As a result, we successfully generated 2D cells from 23 samples (100%), LCOs from 20 samples (87%), and PDXs from 10 samples (43%) (Fig. 1d and Table 1). There was no difference in the success rate of organoid culture among different lung cancer subtypes (Fig. 1d). Establishment of LCOs and PDXs took ~4 weeks and 3–6 months, respectively. Notably, H&E staining showed that LCOs had similar morphological features to patient tissues (Fig. 2). Using this protocol, we banked 80 LCO lines, including the 20 LCOs generated above, from five subtypes of lung cancer: adenocarcinoma, squamous cell carcinoma, adenosquamous carcinoma, large cell carcinoma, and small cell carcinoma (Fig. 1e and Table 2). One of key factors of the organoid biobank will be efficient reconstitution of cryopreserved organoids. Therefore, we performed thawing test for cryopreserved organoids in our biobank. Overall, 39 (70%) of 56 LCOs successfully reconstituted

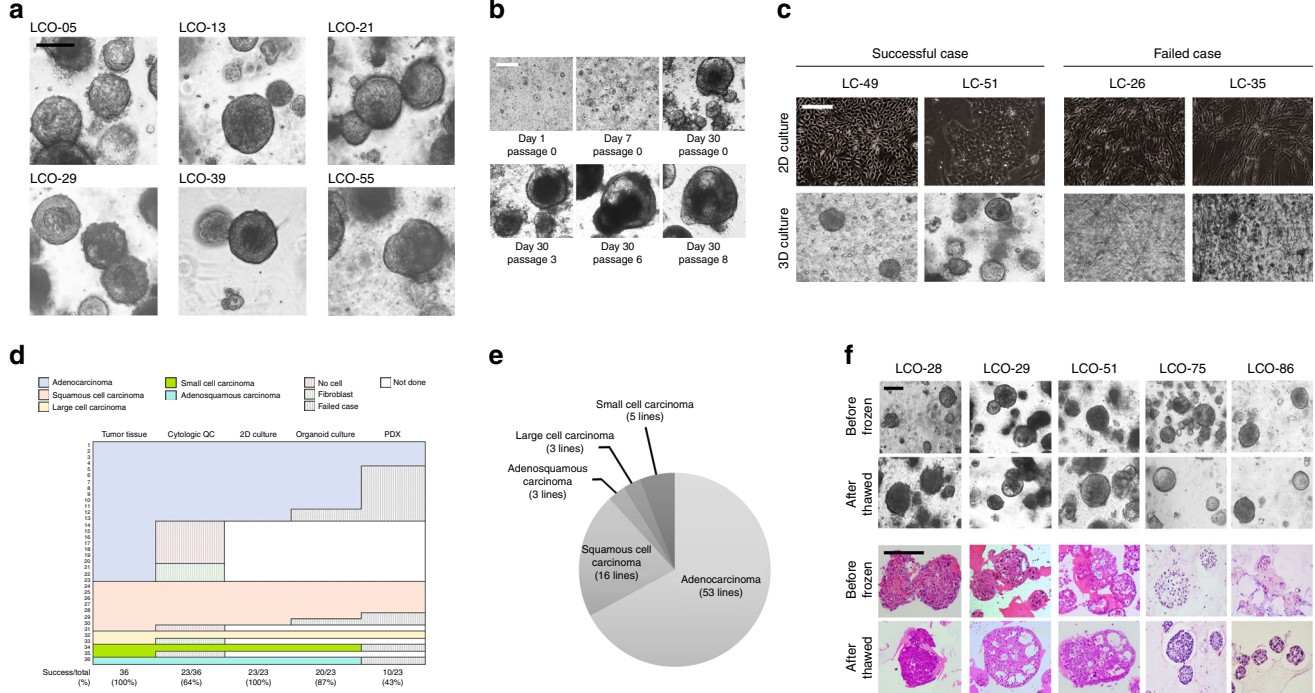

**Fig. 1** LCOs established for the lung cancer biobank. **a** Bright-field microscopy images of LCOs cultured for 2 weeks. Scale bar, 100 μm. The information of LCOs in these images; LCO-05, LCO-36, and LCO-55; adenocarcinoma, LCO-13; squamous cell carcinoma, LCO-21; small cell carcinoma, LCO-29; large cell carcinoma. **b** Representative images of long-term cultured LCOs. This LCO was derived from LC-49. Scale bar, 200 μm. **c** Representative images of successful and failed 2D and 3D cultures derived from lung cancers with different tissue composition. Scale bar, 200 μm. **d** The graph showing the successful or failed cases according to cancer tissue quality and the establishment rate of each cancer models according to lung cancer subtypes. Cytologic QC, cytologic quality check. **e** Pie chart showing the subtypes of established 80 LCOs for the lung cancer biobank. The information of 80 LCOs is shown in Table 2. **f** Bright-field microscopy images and H&E staining images of LCOs before freezing and after thawing. Scale bar, 200 μm. The information of LCOs in these images: LCO-28 — squamous cell carcinoma; LCO-29 — large cell carcinoma; LCO-51 — adenocarcinoma; LCO-75 — small cell carcinoma; LCO-86 — adenosquamous carcinoma

organoid morphology and histologic features of original tissues in our 3D culture protocol (Fig. 1f, Tables 2 and 3).

Taken together, we demonstrated that our culture protocol for LCOs is an efficient and reproducible method for establishing a lung cancer organoid biobank.

**LCOs maintain the histology of original cancer tissues**. It is essential that LCOs sufficiently represent the diverse histological features of different lung cancer subtypes. To compare the morphological and histological features of LCOs with their original LC tissues, we performed H&E staining and immunohistochemistry (IHC) analysis. Similar to their parental cancer tissue, LCOs derived from adenocarcinoma produced acinar or large glandular patterns and retain the expression of lung adenocarcinoma markers[28] napsin-A, thyroid transcription factor 1 (TTF-1), and cytokeratin 7 (CK7) (Fig. 2a). Squamous cell LCOs showed distinct cell borders and cytoplasmic keratinisation, which are histologic characteristics of squamous cell carcinoma tissue[28] (Fig. 2b). They also highly expressed p63 and CK5/6, which are characteristic markers of squamous cell carcinoma[28] (Fig. 2b). Adenosquamous LCOs retained the mixed histological features of adenocarcinoma and squamous cell carcinoma seen in the original patient tumour tissue. They also recapitulated the marker expression pattern of the original cancer tissues: most CK7-expressing organoids derived from these tumours did not show p63 and CK5/6 expression, whereas those that are expressing p63 and CK5/6 did not express CK7 (Fig. 2c). In addition, we investigated whether heterogeneous populations of these adenosqumaous organoids were originated from single cells. In single organoid analysis, adenosquamous carcinoma organoids showed

heterogeneous features. Some organoids were a mixture of p63+ cells and p63− cells. Some organoids were only CK7+, other organoids were only CK5/6+, and others comprised a mixture of CK7+ and CK5/6+ cells, with distinct spatial patterning of cell types (Fig. 2e). In single cell analysis, the organoid like structure formed from a single cell seems to be composed of both adenocarcinoma and squamous carcinoma components (Fig. 2e). However, only few single cells grow to form organoid or organoid-like structure and most wells seeded by a single cell failed to grow to an organoid. This is expected, as many of the cells in an organoid are terminally differentiated and not able to form organoid. This single-cell experiment demonstrates that the adenosquamous phenotype can arise clonally.

Large cell neuroendocrine carcinoma organoids produced a peripheral palisading pattern displayed in large cell neuroendocrine carcinoma tissues and maintained the expression of CK7[34] and a stem cell marker CD133[35,36], but failed to recapitulate expression of neuroendocrine marker CD56[34] and synaptophysin in our culture condition. Small cell lung carcinoma organoids displayed typical small cell morphology of densely packed small round tumour cells with scanty cytoplasm and granular nuclei[28], and diagnostic markers of small cell carcinoma seen in the corresponding cancer tissues, such as CD56[37], synaptophysin[37], and TTF-1 expression[28] (Fig. 2d).

**LCOs retain genetic characteristics of cancer tissues**. To analyse whether LCOs maintain the genetic alterations of their original cancer tissues, we first confirmed that all LCOs match the parental cancer tissues by genetic fingerprinting analysis through SNP genotyping for 24 known SNPs using the MassARRAY

**Table 1 The list of lung cancer samples used for initial cytologic quality check, 2D cell culture, organoid culture and PDX**

| Case | Patient sample | Sex | Age | Lung cancer type | Differentiation | Subtype | Cytologic quality check | 2D-culture | Organoid culture | PDX |
|---|---|---|---|---|---|---|---|---|---|---|
| 1 | LC-05 | F | 61 | Adenocarcinoma | Moderate | Papillary | Epithelial cell-like | + | + | − |
| 2 | LC-10 | F | 63 | Adenocarcinoma | Moderate | Papillary | Fibroblast-like | − | − | − |
| 3 | LC-13 | M | 75 | Squamous cell carcinoma | Moderate | | Epithelial cell-like | + | + | − |
| 4 | LC-15 | F | 75 | Adenocarcinoma | Moderate | Papillary | Epithelial cell-like | + | + | − |
| 5 | LC-17 | M | 70 | Large cell carcinoma | | | Fibroblast-like | − | − | − |
| 6 | LC-18 | M | 59 | Adenocarcinoma | | Foetal | Fibroblast-like | − | − | − |
| 7 | LC-21 | F | 74 | Small cell carcinoma | | | Epithelial cell-like | + | + | − |
| 8 | LC-22 | F | 55 | Adenocarcinoma | Poor | Micropapillary | Epithelial cell-like | + | + | − |
| 9 | LC-23 | M | 83 | Adenocarcinoma | Poor | Solid | No cell | − | − | − |
| 10 | LC-26 | F | 65 | Adenocarcinoma | Moderate | Papillary | Fibroblast-like | − | + | − |
| 11 | LC-28 | M | 78 | Squamous cell carcinoma | Moderate | | Epithelial cell-like | + | + | + |
| 12 | LC-29 | M | 57 | Large cell neuroendocrine carcinoma | Moderate | | Epithelial cell-like | + | + | + |
| 13 | LC-30 | M | 66 | Squamous cell carcinoma | Moderate | | Epithelial cell-like | + | + | + |
| 14 | LC-34 | F | 64 | Adenocarcinoma | Moderate | Acinar | Epithelial cell-like | + | + | + |
| 15 | LC-35 | F | 77 | Adenocarcinoma | Well | Mucinous | Fibroblast-like | − | − | − |
| 16 | LC-36 | M | 68 | Adenocarcinoma | Well | Lepidic | Epithelial cell-like | − | + | − |
| 17 | LC-37 | F | 54 | Adenocarcinoma | Moderate | Papillary | Fibroblast-like | − | − | − |
| 18 | LC-38 | M | 74 | Adenosquamous carcinoma | Moderate | | Epithelial cell-like | + | + | − |
| 19 | LC-39 | M | 75 | Adenocarcinoma | Poor | Solid | Fibroblast-like | − | − | − |
| 20 | LC-40 | M | 81 | Adenocarcinoma | Moderate | Acinar | No cell | − | − | − |
| 21 | LC-41 | M | 48 | Adenocarcinoma | Moderate | Papillary | Fibroblast-like | − | − | − |
| 22 | LC-42 | M | 82 | Squamous cell carcinoma | Moderate | | Epithelial cell-like | + | + | + |
| 23 | LC-43 | F | 71 | Adenocarcinoma | Poor | Solid | Epithelial cell-like | + | + | − |
| 24 | LC-46 | M | 71 | Combined small cell and large cell carcinoma | | | No cell | − | − | − |
| 25 | LC-47 | M | 72 | Squamous cell carcinoma | Moderate | Papillary | No cell | − | − | − |
| 26 | LC-48 | F | 72 | Adenocarcinoma | Moderate | Solid | No cell | − | − | − |
| 27 | LC-49 | M | 75 | Adenocarcinoma | Poor | | Epithelial cell-like | + | + | + |
| 28 | LC-50 | M | 62 | Squamous cell carcinoma | Moderate | | Epithelial cell-like | + | + | + |
| 29 | LC-51 | M | 60 | Adenocarcinoma | Poor | Micropapillary | Epithelial cell-like | + | + | − |
| 30 | LC-52 | M | 72 | Squamous cell carcinoma | Moderate | | Epithelial cell-like | + | + | − |
| 31 | LC-53 | M | 52 | Adenocarcinoma | Moderate | Acinar | Epithelial cell-like | + | − | − |
| 32 | LC-54 | M | 54 | Adenocarcinoma | Moderate | Papillary | Epithelial cell-like | + | + | − |
| 33 | LC-55 | M | 70 | Adenocarcinoma | Poor | Solid | Epithelial cell-like | + | + | − |
| 34 | LC-56 | M | 69 | Squamous cell carcinoma | Well | | Epithelial cell-like | + | + | + |
| 35 | LC-57 | M | 69 | Adenocarcinoma | Well | Mucinous | Epithelial cell-like | + | − | − |
| 36 | LC-58 | M | 63 | Adenocarcinoma | Poor | Solid | Epithelial cell-like | + | + | + |

+ Success, − Fail

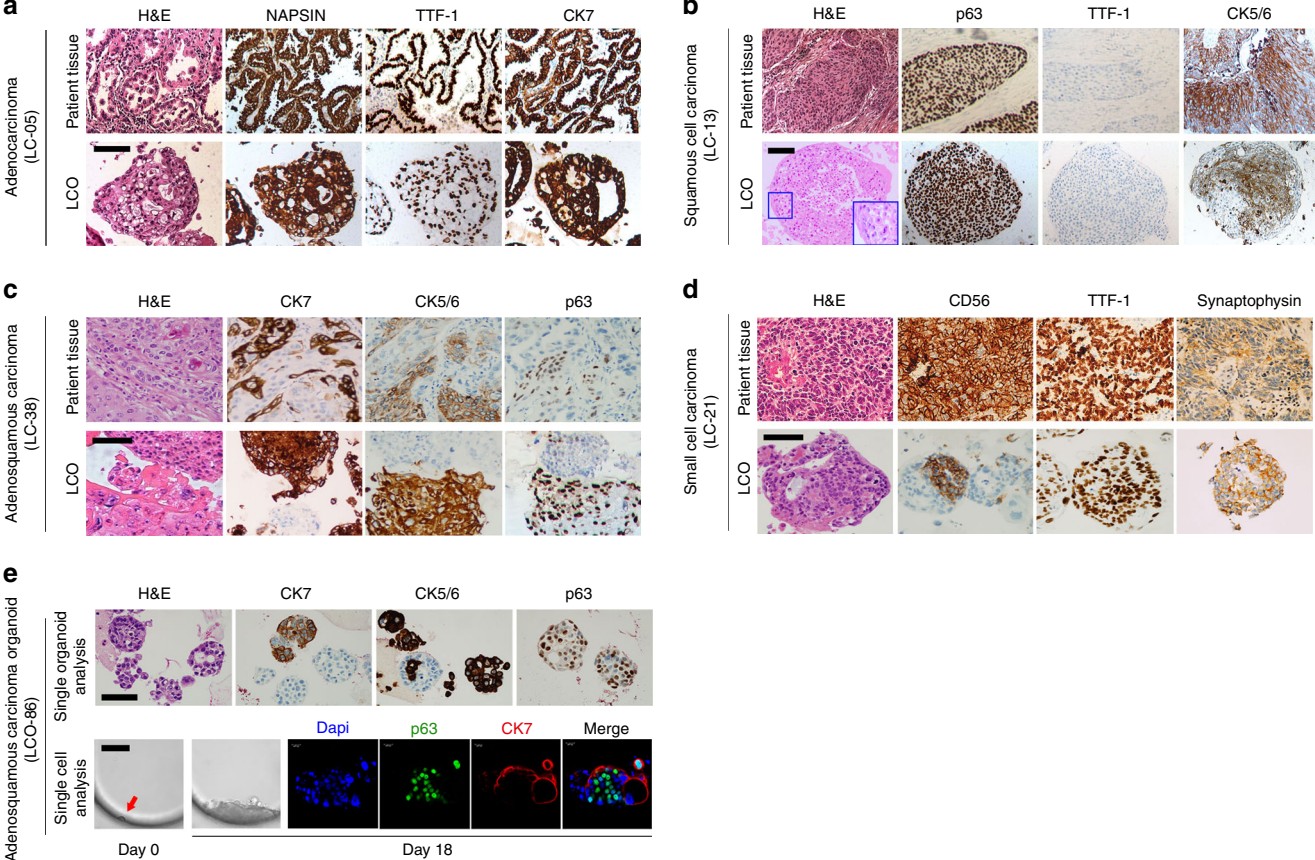

**Fig. 2** LCOs recapitulate the characteristics of the original tissues. **a–d** H&E-stained and IHC-stained images of LCOs and their original LC tissues. The enlarged images in blue boxes of **b** showed cytoplasmic keratinisation of individual squamous carcinoma cells. Scale bars, 100 μm. **e** H&E and IHC-stained images of an adenosquamous carcinoma organoid (upper panel). Individual organoids expressed either adenocarcinoma marker (CK7) or squamous cell carcinoma markers (CK5/6 or p63). Some single organoids showed mixed cell pattern composed of CK7+ and CK7− cells, CK5/6+ and CK5/6− cells, and p63+ and p63− cells. Scale bar, 100 μm. Bright field microscopy and immunofluorescence (IF) images of an adenosquamous carcinoma organoid originated from a single cell (lower panel). The immunofluorescence staining was performed at Day 18 after seeding. The organoid from a single cell was composed of p63+/CK7− cells and p63−/CK7+ cells. The red arrow indicated a single cell seeded in a micro-well. Scale bar in bright-field microscopy images, 100 μm. Scale bar in IF images, 20 μm

platform (Sequenom, San Diego, CA) and found 100% SNP concordance. Next, we performed hybrid capture-based targeted sequencing covering 164 cancer-related genes in 12 LCOs, matched LC tissues, and matched normal tissue. In total, 76 somatic non-synonymous mutations in 44 genes were detected in LC tissues with 77% concordance of somatic mutations in the matched organoids. Furthermore, we compared mutated genes detected in both LC tissues and LCOs for each sample. Most somatic mutations in LC tissues were maintained in their LCOs, as concordance of somatic mutations ranged from 73% to 100% for 11 samples, with the exception of the organoid from LC-30, which had 0% concordance (Fig. 3a, Supplementary Fig. 3a, b), while 100% concordance in SNP genotyping. Additionally, we performed whole exome sequencing (WES) analysis in seven pairs of primary LC tissue, non-neoplastic tissue, and matched LCOs. Median depths of sequencing reads after mapping in human reference genome were 264, 93 and 281 in primary LC tissue, non-neoplastic tissue and matched LCOs, respectively. LCOs retained most variants including driver mutations such as *TP53* and *EGFR* of their original cancer tissues (Fig. 3c, d). The variant allele fraction (VAF) distribution in these organoids generally maintained the VAF distribution in original cancer tissues (Fig. 3d). However, in three cases, relatively less overlap was present between primary cancer tissues and cancer organoids (Fig. 3d). These cases showed a greater number of mutations in

organoids than tissues. This suggests that the organoid-specific mutations were likely present in the originating tumours at low frequencies, rather than being genuinely novel[38] because primary tumour tissue contains normal cells, which lower its VAF (Fig. 3d).

Next, we investigated whether LCOs maintained genetic alterations of original tissues over long-term expansion through WES in five pairs of LCOs in early passage (<passage 4) and late passage (>passage 10). Median depths of sequencing reads were 281 and 268 in early passage organoid and late passage organoids, respectively. Major cancer driver genes and VAF distributions seen at early passage were retained in later passages (Fig. 3e, f). In 4 of 5 cases, the number of mutations increased at later passages, which may suggest sub-clonal expansion (Fig. 3f). Furthermore, most copy number alterations (CNAs) were preserved in late passage organoids and overall patterns of copy number variations were similar between early passage organoid and late passage organoids (Supplementary Fig. 3d). As a result, mutations and CNV detected in original tissues and early passage organoids were maintained during long-term culture.

We measured intratumoural heterogeneity by calculating VAF for each mutated gene[39,40] in LCOs and the matched primary lung cancers. As predicted, VAF values of somatic mutations in LCOs correlated with VAF values in the original tissues, but were higher (Fig. 3b). We suspect that this is due to the non-tumour

**Table 2 The list of lung cancer organoids and normal bronchial organoids in the orgnoid biobank**

| Case | Organoid ID | Sex | Age | Lung cancer type | Differentiation | Subtype | Thawing test |
|---|---|---|---|---|---|---|---|
| 1 | LCO-5 | F | 61 | Adenocarcinoma | Moderate | Papillary | nd |
| 2 | LCO-13 | M | 75 | Squamous cell carcinoma | Moderate | | o |
| 3 | LCO-15 | F | 75 | Adenocarcinoma | Moderate | Papillary | o |
| 4 | LCO-21 | F | 74 | Small cell carcinoma | | | o |
| 5 | LCO-22 | F | 55 | Adenocarcinoma | Poor | Micropapillary | x |
| 6 | LCO-28 | M | 78 | Squamous cell carcinoma | Moderate | | o |
| 7 | LCO-29 | M | 57 | Large cell neuroendocrine carcinoma | | | o |
| 8 | LCO-30 | M | 66 | Squamous cell carcinoma | Moderate | | x |
| 9 | LCO-34 | F | 64 | Adenocarcinoma | Moderate | Acinar | nd |
| 10 | LCO-36 | M | 68 | Adenocarcinoma | Well | Lepidic | nd |
| 11 | LCO-38 | M | 74 | Adenosquamous carcinoma | | | nd |
| 12 | LCO-42 | M | 82 | Squamous cell carcinoma | Moderate | | o |
| 13 | LCO-43 | M | 71 | Adenocarcinoma | Poor | Solid | o |
| 14 | LCO-49 | M | 75 | Adenocarcinoma | Poor | Solid | o |
| 15 | LCO-50 | M | 62 | Squamous cell carcinoma | Moderate | | o |
| 16 | LCO-51 | M | 60 | Adenocarcinoma | Poor | Micropapillary | o |
| 17 | LCO-52 | M | 72 | Squamous cell carcinoma | Moderate | | o |
| 18 | LCO-55 | M | 70 | Adenocarcinoma | Poor | Solid | o |
| 19 | LCO-56 | M | 69 | Adenocarcinoma | Well | Mucinous | x |
| 20 | LCO-58 | M | 63 | Adenocarcinoma | Poor | Solid | o |
| 21 | LCO-59 | M | 72 | Adenocarcinoma | Poor | Solid | nd |
| 22 | LCO-60 | M | 57 | Adenocarcinoma | Poor | Acinar | nd |
| 23 | LCO-61 | M | 66 | Adenocarcinoma | Well | Mucinous | x |
| 24 | LCO-62 | M | 63 | Adenocarcinoma | Moderate | Papillary | o |
| 25 | LCO-63 | M | 66 | Adenocarcinoma | Poor | Solid | nd |
| 26 | LCO-64 | F | 60 | Adenocarcinoma | Moderate | Acinar | nd |
| 27 | LCO-65 | M | 72 | Adenocarcinoma | Moderate | Acinar | nd |
| 28 | LCO-66 | M | 49 | Adenocarcinoma | Moderate | Acinar | nd |
| 29 | LCO-67 | F | 59 | Adenocarcinoma | Moderate | Enteric | nd |
| 30 | LCO-68 | M | 72 | Squamous cell carcinoma | Moderate | | nd |
| 31 | LCO-69 | F | 64 | Adenocarcinoma | Well | Lepidic | nd |
| 32 | LCO-70 | M | 64 | Adenocarcinoma | Poor | Sarcomatoid** | o |
| 33 | LCO-71 | M | 54 | Adenocarcinoma | Poor | Acinar | x |
| 34 | LCO-72 | M | 47 | Squamous cell carcinoma | Moderate | | nd |
| 35 | LCO-73 | F | 70 | Adenocarcinoma | Moderate | Acinar | nd |
| 36 | LCO-74 | M | 75 | Squamous cell carcinoma | Moderate | | o |
| 37 | LCO-75 | M | 69 | Small cell carcinoma | | | o |
| 38 | LCO-76 | M | 61 | Adenocarcinoma | Moderate | Papillary | nd |
| 39 | LCO-77 | M | 67 | Adenocarcinoma | Moderate | Papillary | nd |
| 40 | LCO-78 | M | 67 | Adenocarcinoma | Poor | Solid | nd |
| 41 | LCO-79 | F | 62 | Adenocarcinoma | Moderate | Mucinous | x |
| 42 | LCO-80 | M | 82 | Squamous cell carcinoma | Moderate | | o |
| 43 | LCO-81 | M | 64 | Adenocarcinoma | Poor | Solid | x |
| 44 | LCO-82 | M | 60 | Squamous cell carcinoma | Moderate | | nd |
| 45 | LCO-83 | M | 65 | Squamous cell carcinoma | Moderate | | nd |
| 46 | LCO-84 | M | 63 | Small cell carcinoma | | | o |
| 47 | LCO-85 | F | 64 | Adenocarcinoma | Moderate | Acinar | x |
| 48 | LCO-86 | M | 77 | Adenosquamous carcinoma | | | o |
| 49 | LCO-87 | M | 63 | Adenocarcinoma | Moderate | Papillary | o |
| 50 | LCO-88 | M | 76 | Large cell neuroendocrine carcinoma | | | x |
| 51 | NBO-89 | M | 50 | Normal bronchus | | | o |
| 52 | NBO-90 | F | 77 | Normal bronchus | | | o |
| 53 | LCO-91 | M | 65 | Small cell carcinoma | | | o |
| 54 | NBO-92 | F | 63 | Normal bronchus | | | nd |
| 55 | NBO-93 | M | 54 | Normal bronchus | | | nd |
| 56 | NBO-94 | M | 70 | Normal bronchus | | | nd |
| 57 | LCO-95 | M | 57 | Squamous cell carcinoma | Moderate | | x |
| 58 | LCO-96 | F | 71 | Adenocarcinoma | Moderate | Metastatic* | o |
| 59 | LCO-97 | M | 64 | Adenocarcinoma | Moderate | Papillary | x |
| 60 | LCO-98 | M | 75 | Squamous cell carcinoma | Moderate | | x |
| 61 | LCO-99 | M | 59 | Adenocarcinoma | Poor | Solid | o |
| 62 | LCO-100 | M | 58 | Adenocarcinoma | Moderate | Acinar | x |
| 63 | LCO-101 | F | 45 | Adenocarcinoma | Moderate | Acinar | nd |
| 64 | LCO-102 | M | 70 | Adenocarcinoma | Moderate | Acinar | nd |
| 65 | LCO-103 | M | 76 | Adenocarcinoma | Moderate | Acinar | nd |
| 66 | LCO-104 | M | 62 | Adenocarcinoma | Poor | Sarcomatoid** | x |
| 67 | LCO-105 | F | 68 | Adenocarcinoma | Moderate | Papillary | nd |
| 68 | LCO-106 | M | 67 | Adenocarcinoma | Moderate | Mucinous | o |
| 69 | LCO-107 | F | 62 | Adenosquamous carcinoma | | | x |
| 70 | LCO-108 | M | 63 | Adenocarcinoma | Moderate | Acinar | nd |
| 71 | LCO-109 | F | 52 | Squamous cell carcinoma | Moderate | | o |
| 72 | LCO-110 | M | 52 | Adenocarcinoma | Moderate | Metastatic* | o |
| 73 | LCO-111 | M | 70 | Large cell neuroendocrine carcinoma | | | o |
| 74 | LCO-112 | M | 60 | Adenocarcinoma | Moderate | Metastatic* | o |
| 75 | LCO-113 | M | 62 | Adenocarcinoma | Moderate | Mucinous | o |
| 76 | LCO-114 | M | 64 | Small cell carcinoma | | | o |
| 77 | LCO-115 | M | 77 | Adenocarcinoma | Moderate | Acinar | x |
| 78 | LCO-116 | M | 80 | Squamous cell carcinoma | Moderate | | o |
| 79 | LCO-117 | M | 63 | Adenocarcinoma | Moderate | Metastatic* | o |
| 80 | LCO-118 | M | 57 | Adenocarcinoma | Moderate | Papillary | x |
| 81 | LCO-119 | M | 63 | Adenocarcinoma | Moderate | Acinar | o |
| 82 | LCO-120 | M | 59 | Adenocarcinoma | Moderate | Metastatic* | o |
| 83 | LCO-121 | M | 71 | Adenocarcinoma | Moderate | Papillary | o |
| 84 | LCO-122 | M | 81 | Adenocarcinoma | Moderate | Metastatic* | o |
| 85 | LCO-123 | M | 42 | Adenocarcinoma | Moderate | Metastatic** | nd |

*LCO* lung cancer organoid, *NBO* normal bronchial organoid, *nd* not done, *O* pass, *X* fail, *Metastatic** metastatic colonic adenocarcinoma, *Metastatic*** metastatic lung adenocarcinoma, biopsy, *Sarcomatoid*** adenocarcinoma with sarcomatoid feature

**Table 3 Reconstitution of the cryopreserved organoids according to histologic subtypes**

| Sample subtypes | Reconstitution success no. (%) | Reconstitution fail no. (%) |
|---|---|---|
| Adenocarcinoma | 20 (51) | 12 (31) |
| Squamous cell carcinoma | 9 (23) | 3 (8) |
| Large cell carcinoma | 2 (5) | 1 (3) |
| Small cell carcinoma | 5 (13) | 0 (0) |
| Adenosquamous carcinoma | 1 (3) | 1 (3) |
| Normal bronchial organoid | 2 (5) | 0 (0) |
| Total (56 samples) | 39 (70) | 17 (30) |

stroma and immune infiltrates contained in lung cancer specimens. Organoids were grown from isolated epithelial populations without any stromal support or immune infiltrates[40]. Therefore, if a mutation is common in all cancer cells, it will be represented in all sequencing data and thereby reaching to the highest score (=1). Nevertheless, VAF values in LCOs widely ranged between 0.1 and 1.0 (Fig. 3b and Supplementary Fig. 3c), suggesting that LCOs maintain genetic heterogeneity with both major and minor subclonal cancer cell populations. In summary, we demonstrated that LCOs maintained the genomic alterations and genomic heterogeneity of their original cancer tissues.

**NBOs maintain bronchial epithelial organisation**. When assessing carcinogenesis and drug toxicity, it is important to have a normal tissue control. To address this, we dissected non-neoplastic bronchial mucosa adjacent to the cancer tissue and cultured under the same culture conditions as LCOs using suboptimal MBM. At first, these normal bronchial cells formed organoids; during subsequent subcultures, however, they failed to form organoids (Fig. 4a). The induction of Wnt signalling and inhibition of BMP/TGFβ signalling promote tissue renewal[41]; therefore, we added the following supplements for lung development into medium[14]: Noggin (BMP and TGFβ signalling blocker) and A83-01 (an activin receptor-like kinase (ALK) inhibitor) to inhibit BMP and the TGFβ signalling pathway, and Wnt3a to induce Wnt signalling. With these supplements, organoids successfully formed from the bronchial cells over 10 passages (Fig. 4a). In this condition, our NBOs had a round shape similar to previously reported normal lung organoids[31]. When these organoids grew, budding tubule-like organoids appeared starting in passage 4, and maintained the morphology for subsequent long-term expansion over 10 passages (Fig. 4a). H&E data demonstrated that these organoids have pseudostratified epithelium composed of basal cells and luminal cells including secretory and ciliated cells, as in normal bronchial mucosa tissue[42] (Fig. 4b). Immunofluorescence (IF) showed that the organoids were composed of basal cells expressing p63 in the outer cell layer[25,28], as well as mucin-secreting cells (goblet cells) expressing MUC1[28] and ciliated cells expressing acetylated-tubulin and Arl13b in the luminal cell layer[25,28] (Fig. 4c). Moreover, some organoids included club cells expressing CC10[43] (Fig. 4c). However, the distribution of cell types in NBOs differed from those in of LCOs. Adenocarcinoma organoids had only MUC1-positive cells with few ciliated cells (Fig. 4d), and squamous cell carcinoma organoids were consisted of p63-positive cells with few ciliated cells (Fig. 4e). Interestingly, the type of cilia in NBOs and LCOs was different; NBOs had motile cilia found in large bundles[44], on the luminal surface of inner layer cells, but cells in LCOs had primary cilia found as a single appendage per cell[44] (yellow boxes in Fig. 4c–e). Thus, we demonstrated that

NBOs represent the spatial arrangement of normal bronchial mucosa.

**LCOs maintain tumourigenic character**. To evaluate the tumourigenicity of LCOs, we transplanted LCOs and 2D-LC cells that originated from the same LC tissues into immunodeficient mice. Since LCOs have 3D structures (Fig. 1), we developed a new graft method for maintaining this 3D structure upon engrafting (Supplementary Fig. 4). Instead of injecting dissociated cancer cells from organoids as previously described[19,22,45], we cast the LCOs on a cellulose membrane using Matrigel and grafted them subcutaneously in immunodeficient mice (Supplementary Fig. 4). All of the transplanted LCOs (6/6) produced tumours in the mice, whereas only 2 out of 6 2D-LC cells did (Fig. 5a, b). Furthermore, LCOs showed higher tumour take rates in mice (6/6) than direct grafts of the original cancer tissues (2/6) (Table 4). In addition, we compared our new method of xenografting whole organoids with existing methods[19,22,45]. Consistent with previously described methods, we dissociated organoids into single cells and subcutaneously injected them into immunodeficient mice. In parallel using our new method, the same number of LCOs were transplanted whole into the same mice. Transplanted 3D LCOs formed tumours in mice more successfully (6/6 vs. 3/6) with bigger tumours (Supplementary Fig. 5) and in a shorter time period, excepting LCO-43 in which dissociated cells formed tumour faster than 3D organoids (Fig. 5c). Although the LCO-43 formed tumour later than dissociated cells, final sizes of tumours were similar in a same time period (Supplementary Fig. 5a). Therefore, it can be considered that maintaining the 3D structure of organoids may have advantage in tumour growth in a xenograft condition to improve the efficiency of xenograft formation. We also assessed morphology of the xenograft tumours formed from LCOs by comparison with the morphology of their parental LC tissues and LCOs. In H&E staining, they showed high similarity with the original LC tissues and LCOs (Supplementary Fig. 6).

**LCOs maintain expression of PD-L1**. One of the emerging hallmarks of cancer is immune evasion, in which cancer evades the anti-tumour responses of the immune system[46,47]. One mechanism for immune evasion is the expression of immune checkpoint molecules, such as PD-L1 by tumour cells, and targeting these checkpoint molecules with antibodies is emerging as an effective therapeutic strategy against lung cancer[3,48,49]. We tested the LCOs and PDXs for PD-L1 expression by IHC and compared to original tumour tissues. Two out of 10 original tumour tissues expressed PD-L1, and the same pattern was observed in matching LCOs and PDXs (Supplementary Fig. 6b).

**LCOs for in vitro patient-specific drug trials**. To assess whether our LCOs were useful for in vitro drug sensitivity testing, we generated dose-response curves for several drugs and calculated half-maximal inhibitory concentrations ($IC_{50}$). Docetaxel, targeting cellular microtubules[50], induced cell death in various LCOs as well as a NBOs with different sensitivities. Notably, NBO is remarkably vulnerable to the cytotoxic drug docetaxel ($IC_{50} = 0.08\ \mu M$) comparing to targeted drugs, Olaparib ($IC_{50} = 69\ \mu M$), Erlotinib ($IC_{50} > 100\ \mu M$), and Crizotinib ($IC_{50} = 3\ \mu M$) (Fig. 6a–c, f).

We next tested drugs that target the specific genetic alterations identified in different LC tissues. The PARP inhibitor olaparib targets tumours with homologous recombination repair deficiencies, including those with mutations in the DNA repair encoding gene *BRCA2*[51–53]. Two LCOs, LCO-28 and LCO-55, harboured different *BRCA2* alterations and responded differently to treatment with olaparib. LCO-55 had a BRCA2 p.W2619C mutation

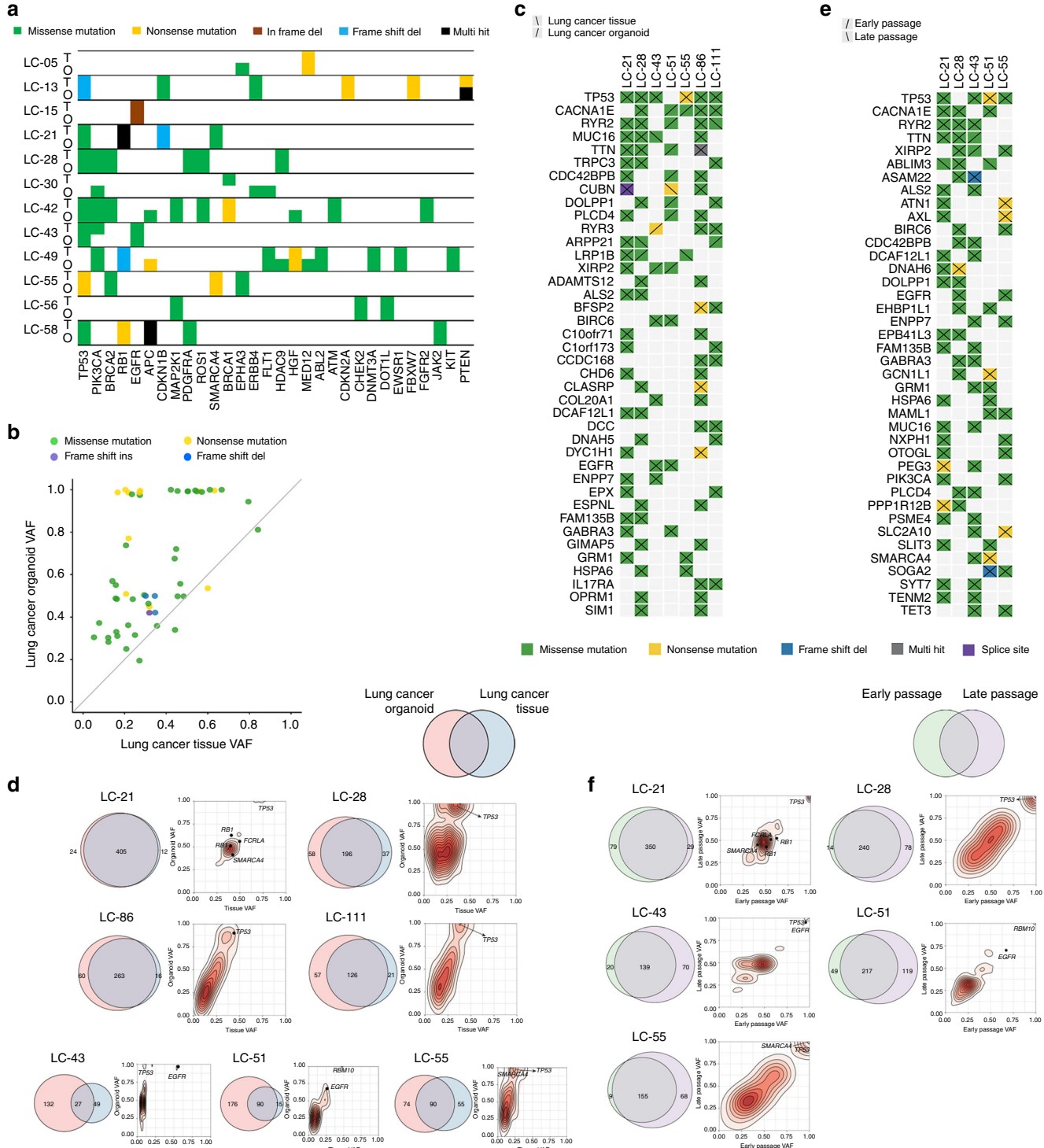

**Fig. 3** LCOs retain the genetic characteristics of the original tissues after long-term cultured. **a** Heat-map analysis of the top 30 mutations in LCOs and their corresponding LC tissues. O LCO, T patient tissue. **b** Comparison of VAF of genetic alterations detected in LCOs and LC tissues. **c** Heatmap showing somatic mutations affecting cancer genes in LC tissues and paired LCOs. **d** Venn diagrams indicating the number of somatic mutations present in each LC tissues and their paired LCOs. **e** Heatmap showing somatic mutations affecting cancer genes in early passage and late passage organoids. **f** Venn diagrams indicating the number of somatic mutations present in early passage and late passage organoids

and a lower olaparib IC$_{50}$ than LCO-28, which had a BRCA2 p.M965I mutation (Fig. 6b) and the structure of LCO-55 was destroyed by olaparib treatment (Supplementary Fig. 7). The tumour growth of PDX from LCO-55 was also inhibited after olaparib treatment (Supplementary Fig. 8a). Protein function prediction using PolyPhen-2[54] suggested that p.W2619C is

predicted to be detrimental to BRCA2 function, while p.M965I is non-pathogenic to BRCA2 function. Our drug response results agree with the hypothesis that BRCA2 p.W2619C is a pathogenic alteration that exerts synthetic lethality with PARP inhibition. Follow-up experiments are needed to further investigate this hypothesis, as the drug responses are not evidence of mechanism.

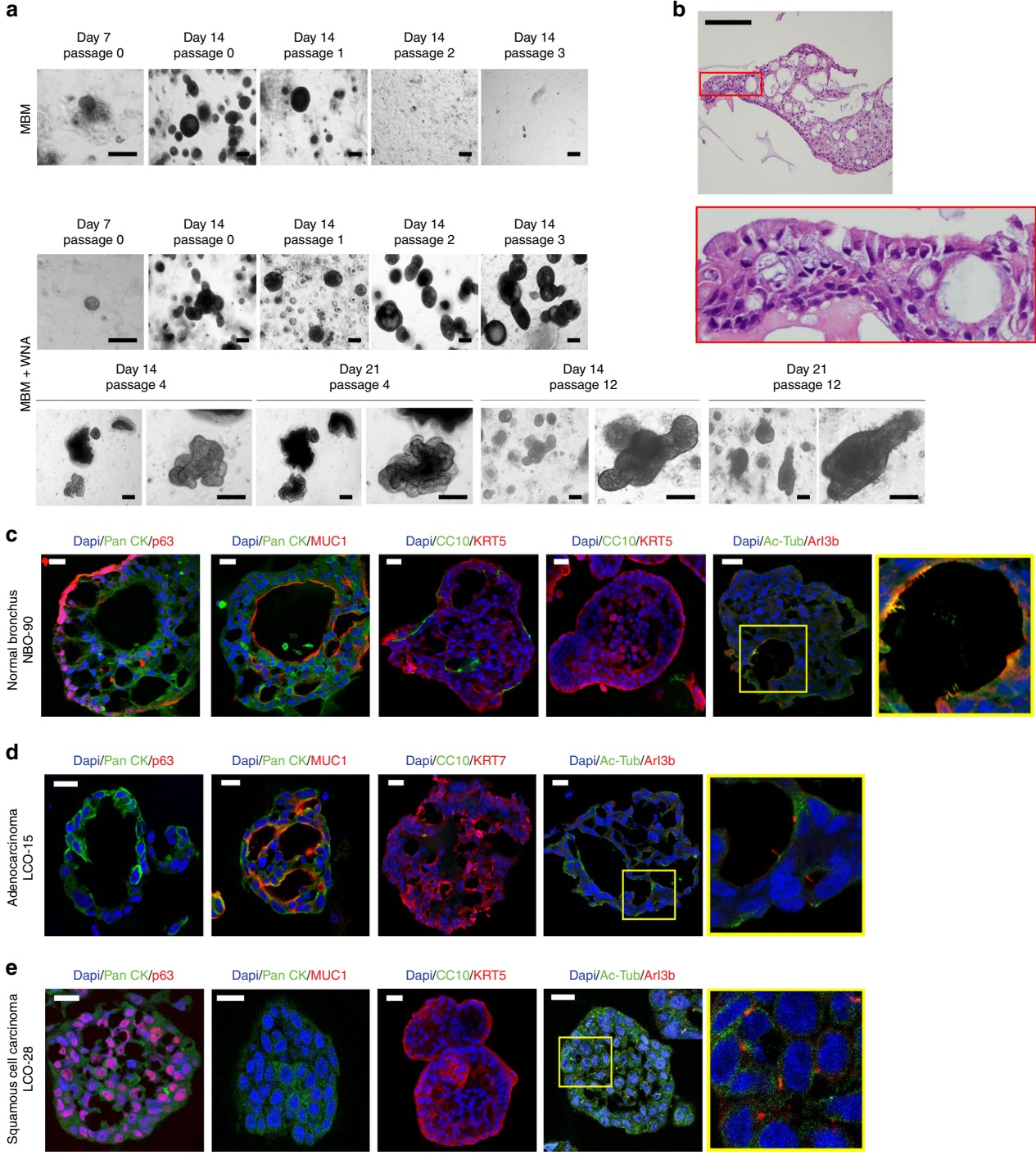

**Fig. 4** Organ-like structures of the NBOs and LCOs. **a** Bright-field microscopy images of a NBO, NBO-90. NBO-90 was cultured in MBM and MBM + WNA (W; Wnt3A, N; Noggin, A; A83-01). Scale bars, 200 μm. **b** H&E staining image of NBO-90. Scale bar, 100 μm. **c–e** Immunofluorescence images showing the expression of pancytokeratin, p63, MUC1, KRT5, KRT7, CC10, and cilia co-stained with acetylated α-tubulin (Ac-Tub) and Arl13b (The enlarged images in yellow boxes showed cilia co-stained with Ac-Tub and Arl13b). Nuclei (blue) were stained with DAPI. Scale bar, 20 μm

EGFR tyrosine kinase inhibitors, such as erlotinib, are the most widely used drugs targeting specific molecular alterations in lung cancer. Two LCOs, LCO-43 and LCO-51, had EGFR p.L858R mutations but showed different responses to erlotinib. Whereas LCO-43 displayed high sensitivity to erlotinib, LCO-51 was resistant to erlotinib, with a similar $IC_{50}$ value to an organoid without EGFR mutations (Fig. 6c and Supplementary Fig. 7). Since our LCOs originated from cancer tissues of drug-naive patients, we hypothesised that this differential response of

LCO-51 to erlotinib could be associated with an intrinsic resistance mechanism. Based on copy number analysis of genomic data, LCO-51 had amplification of *MET* (Fig. 6d), which is associated with resistance to EGFR tyrosine kinase inhibitors[55]. IHC analysis confirmed strong c-Met protein expression in LC-51 tissues and LCO-51 organoids (Fig. 6e). The c-Met inhibitor crizotinib effectively targeted LCO-51 organoids (Fig. 6f and Supplementary Fig. 7). In the PDX model, drug treatment had similar results. Critzotinib suppressed the

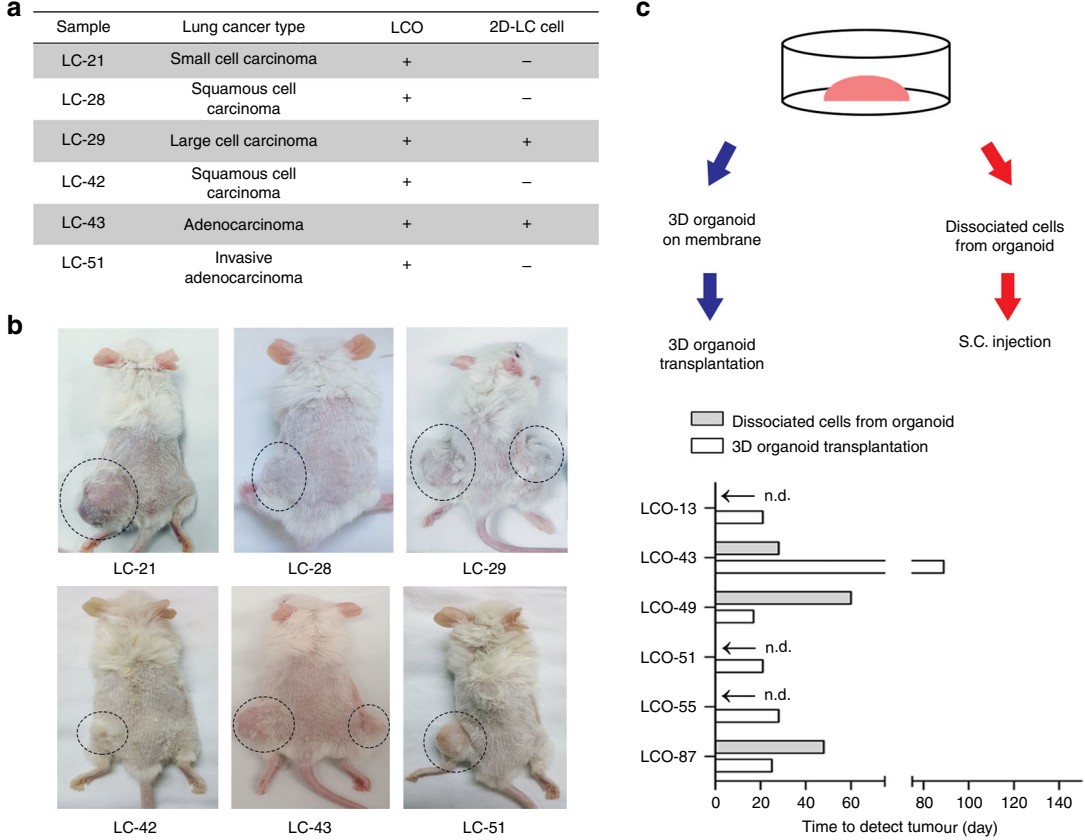

**Fig. 5** PDXs from LCOs maintain the characteristics of the original tissues. **a** List of PDXs established by transplanting LCOs or injecting 2D-LC cells. Transplanted LCOs and injected 2D-LC cells that formed tumours in mice are marked as "+", and the LCOs and 2D-LC cells that did not form tumours are marked as "−". **b** Images of mice used to establish PDXs. LCOs (left) and 2D-LC cells (right) were simultaneously transplanted or injected into the same mouse. Circles indicate the formed tumours. **c** Experimental design and results. Bar graph showing the number of days until tumour formation. n.d: tumour not detected

**Table 4 Comparison of graft success between direct tissue graft and 3D-organoid graft methods**

| Case | Patient sample | Lung cancer subtype | Tissue-PDX | LCO-PDX |
|------|----------------|---------------------|------------|---------|
| 1 | LC-21 | Small cell carcinoma | − | + |
| 2 | LC-28 | Squamous cell carcinoma | − | + |
| 3 | LC-29 | Large cell carcinoma | + | + |
| 4 | LC-42 | Squamous cell carcinoma | + | + |
| 5 | LC-43 | Adenocarcinoma | − | + |
| 6 | LC-51 | Adenocarcinoma | − | + |

*Tissue-PDX* PDX by direct tissue graft method, *LCO-PDX* PDX by 3D lung cancer organoid graft method, + success, − fail

growth of tumour from engrafted LCO-51 more effectively than erlotinib (Supplementary Fig. 8b).

The findings were further supported by analysis of downstream signalling molecules of the receptor tyrosine kinase pathway (Fig. 6g). In LCO-43, mediators of EGFR signalling, including p-EGFR, p-Mek, and p-Erk, were inhibited by erlotinib treatment. However, mediators of c-Met signalling, including p-Akt, p-Mek, and p-Erk, were not decreased by crizotinib treatment (Fig. 6g). In LCO-51, crizotinib still inhibited p-Akt, p-Mek, and p-ERK. However, erlotinib treatment did not decrease expression of p-Mek

and p-Erk in LCO-51, despite a decrease in p-EGFR expression (Fig. 6g). These results demonstrated that our LCOs could be used for predicting patient-specific drug responses in vitro, as well as proof-of-concept studies for new targeted drugs based on genetic alterations.

## Discussion

In vitro lung cancer models representing individual patients will facilitate the development of personalised medicine for lung cancer. Here, we described a method for successful and efficient generation of LCOs that generally recapitulate the characteristics of the original patient tumours. We established a living biobank of 80 LCOs derived from five subtypes of lung cancer that cover more than 95% of lung cancer patients: adenocarcinoma, squamous cell carcinoma, small cell carcinoma, adenosquamous carcinoma, and large cell carcinoma. LCOs maintained the characteristic histological features of the original cancer tissues. To reduce failure rate, we conducted initial cytologic quality evaluation for tumour purity in tumour tissues. It has been already reported that stromal co-culture for LC tissues can result in fibroblast contamination in vitro[56]. To avoid this problem, we established feeder-free LCO lines at an 87% success rate, post-cytologic quality evaluation. As a patient-derived cancer model, 2D-LC cells are also valuable with higher initial success rate than organoid culture (Fig. 1d). Notably, 2D-LC cells could not be cultured for long term in this 2D culture condition. After passage 5, their growth rate slowed and cells underwent senescence. This

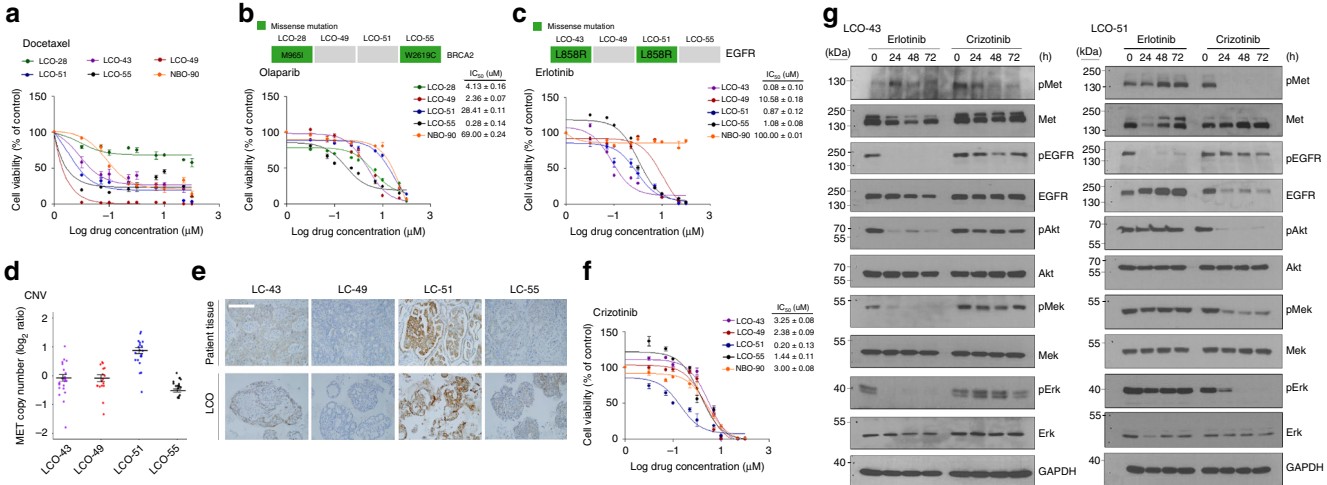

**Fig. 6** LCOs as a platform for predicting therapeutic responses. **a** Dose-response curves after 6 days of treatment with docetaxel. The cell viability was measured by luminescence signal intensities. **b** and **c** Dose-response curves after 6 days of treatment with olaparib **b** or erlotinib **c**. The indicated drug concentrations were used to treat two LCO lines with mutations (green box), another two LCO lines without mutations (grey box) and the NBO. Representative viability curves were generated from luminescence signal intensities. **d** The copy number variation (CNV) graph showing MET copy number amplification in the four LCO lines. **e** IHC-stained images showing c-MET expression in the four patient tissues and LCOs. Scale bar, 100 μm. **f** Dose-response curves after treatment with indicated drug concentrations of crizotinib in four LCOs and a NBO. Representative viability curves were generated from the luminescence signal intensities. **g** Immunoblot analysis showing changed protein expressions mediating EGFR signalling and c-Met signalling transduction after treating erlotinib and crizotinib. Two LCOs, LCO-43 and LCO-51 were treated with 1 μM erlotinib and 1 μM crizotinib for 24, 48 and 72 h. GAPDH was used as a loading control. Full-length blot is shown in the supplementary information, and the gels have been run under the same experimental conditions. All error bars in **a–c** and **f** indicate SEM, n = 3. In **b**, **c** and **f** IC$_{50}$ values are the average ± SD of each condition analysed in triplicate. When error bars are not visible they are smaller than the size of the symbol

limited availability of 2D-LC cells has been shown in other published papers[57]. Under this consideration, LCO models have greater long-term potential than 2D-LC cells. These LCOs maintained the expression patterns of subtype-specific markers. Genomic analysis confirmed that the LCOs maintained up to 77% concordance of the mutations that were present in the original cancer tissues. The VAFs of tissues and LCOs were variable and linearly correlated, indicating that the tumour genetic heterogeneity was maintained in LCOs.

In 3D culture conditions with a commercial matrix (Matrigel), cells undergo organoid formation through epithelial cell interactions[17,58]. As a result, organoids have spatial configurations made by diverse types of epithelial cells, as observed in vivo[17]. As expected, NBOs cannot be cultured in MBM because MBM was a suboptimal media for inhibition of normal cell growth. Based on previous papers regarding culture of normal lung organoids using airway epithelium derived from human lung stem cells and pluripotent stem cells[14,26,28,29], we added Wnt3a, Noggin and A83-01 to MBM for promoting lung development. As a result, normal bronchial cells which were derived from large airway tissues formed organoids that successfully differentiated to the airway epithelium in MBM + WNA (Fig. 4a–c). The previous report showed normal lung organoids having round shapes, which were derived from peripheral conducting airway[31]. However, our normal lung organoids had budding shapes not only round shapes, as cultured over passage 4. We suspect that this morphological difference is associated with the region of the resected tissue and the culture media components, such as growth factors and small molecules. Our NBOs recapitulate cellular composition and spatial organisation of normal bronchial mucosa consisted of p63-positive basal cells, mucin-secreting goblet cells, ciliated respiratory epithelial cells, and club cells expressing CC10 (Fig. 4c). In contrast, LCOs did not undergo this differentiation and consisted largely of a single cell type: mucin-secreting cells in adenocarcinoma or p63-positive cells in squamous cell carcinoma

(Fig. 4d, e). Recently generation of LCOs have been reported as a part of human airway organoids for disease modelling[31]. In this paper, a highly nutritious optimal medium was used for cultivation of both normal airway organoids and LCOs. Outgrowth of normal organoids is an important cause of failure during cultivation of cancer organoids using an optimal medium for normal organoids[23,59]. Therefore, to remove normal organoids, they used manual separation method based on organoids morphology. In case of TP53-mutated cancer samples, they added Nutlin-3a to the culture medium to induce death of p53 wild type normal cells. Manual separation is tedious time consuming work. For using Nutulin-3a selection method, tumour genomic sequencing prior to organoid culture is required. In our study, we describe a well-designed sub-optimal medium for high-efficiency culture of LCOs without being bothered with normal cell contamination and an optimal culture medium for NBOs. As results, we successfully biobanked 80 LCO lines covering five different histological types and five NBOs.

A critical limitation of cancer organoid models is the lack of a cancer microenvironment including stromal cells and immune cells, because cancer organoids are derived only from epithelial cells. Thus, to investigate the interaction between cancer cells and the microenvironment, a co-culture system with immune/stromal cells[30,50] or xenograft models[30] is necessary. In addition, the LCO-derived xenografts maintained the characteristics of their parental organoids and original cancer tissues. Therefore, we expect that xenograft models made from 3D-LCOs would be advantageous not only to generate a large number of PDXs but also to study the in vivo interactions between stromal cells, immune cells, and cancer organoids.

Recently, immune checkpoint inhibitors have been introduced in the clinic, though biomarkers are still being assessed[46,60]. Specifically, PD-L1 expression in tumour cells has been identified as a predictive biomarker for response to PD-1 inhibitors, such as pembrolizumab[48,49]. Tumour cells induce PD-L1 expression by

their intrinsic and/or extrinsic regulation[30]. Although organoid cultures do not include immune cells, LCOs and PDXs from these organoids maintained PD-L1 expression. In future studies, we expect that the use of a co-culture system with tumour-infiltrating lymphocytes and LCOs may be used to assess the efficacy of immune checkpoint inhibitors[61].

Genetic alterations in lung cancer are extraordinarily diverse[62]. Therefore, development of a truly personalised treatment for lung cancer may require genomically annotated individual patient avatars for testing experimental therapeutics and combination therapy. In this study, we showed the potential of LCOs as a new preclinical cancer model representing individual patients. For example, the efficacy of olaparib, which targets cancers with deficiency in homologous recombination repair[53], differed in LCOs harbouring different *BRCA2* alterations of unknown significance. Drug tests in the LCOs also identified potential differences in drug resistance. Although LCO-43 and LCO-51 had the same EGFR mutation, they responded differently to erlotinib and crizotinib due to different secondary alterations.

In the era of personalised cancer medicine, new clinical trial designs for drug repurposing tests, drug combination tests, and n-of-1 trials are becoming increasingly important compared with classic clinical designs involving large and diverse patient populations. Approximately 26% of stage I and II non-small-cell lung cancers (NSCLCs) recur at local regions or distant sites after curative surgery[63]. We showed that biobanked LCOs can be used for anticancer drug screening or for in vitro trials for predicting individual patient drug responses. In this way, LCOs may be useful in comparing efficacies of multiple adjuvant therapies. Using our technology, longitudinal sampling and LCO biobanking from individual patients during clinical course may validate these studies and facilitate the personalisation of cancer treatment, as well as the evaluation of drug toxicity by comparing to normal lung organoids.

In conclusion, we established LCOs that recapitulated the histological and genetic features of the five most common subtypes of lung cancer. The LCOs also maintained genomic heterogeneity from the parental lung cancer. Our studies suggest that the LCO system will be a useful platform for drug screening and new clinical trials. Additionally, our NBOs may be used for estimation of drug toxicity on non-cancerous cells. Considering the short length of time from establishment to drug testing, our model can be used for predicting patient-specific drug responses as well as broader preclinical studies.

## Methods

**Human specimens**. Small pieces (~1–4 cm$^3$) of LC tissues and non-neoplastic bronchial tissues were taken from surgically resected lung specimens as part of the lung cancer biobanking process at the Asan Bio-Resource Center (Seoul, Korea) with patients' informed consent. The research protocol was approved by the Ethics Committee of the Asan Medical Center (Seoul, Korea). The entire experimental protocol was conducted in compliance with the institutional guidelines. Samples were confirmed as tumour or normal tissue on the basis of histopathological assessment. The diagnosis of each case was confirmed by pathologists at Asan Medical Center.

**Tissue preparation and culture of 2D-LC cells, LCOs and NBOs**. Parts of the human samples (5–6 pieces of 1 mm$^3$ tissue) were separated and transported to the laboratory within 1 h of removal from the patients in cold Hank's balanced salt solution (HBSS) with antibiotics (Lonza, Basel, Switzerland). Samples were washed three times with cold HBSS with antibiotics and were sectioned with sterile blades. Approximately two-thirds of the sectioned samples were used to establish PDX models, and the rest were incubated with 0.001% DNase (Sigma-Aldrich, MO, USA), 1 mg ml$^{-1}$ collagenase/dispase (Roche, IN, USA), 200 U ml$^{-1}$ penicillin, 200 mg ml$^{-1}$ streptomycin, and 0.5 mg ml$^{-1}$ amphotericin B (2% antibiotics, Sigma) in DMEM/F12 medium (Lonza) at 37 °C for 2 h with intermittent agitation. After incubation, the suspensions were repeatedly triturated by pipetting and passed through 70-μm cell strainers (BD Falcon, CA, USA). The strained cells were centrifuged at 112 × *g* for 3 min, and the pellet was resuspended in 100 μl MBM (serum-free medium (DMEM/F12; Lonza) supplemented with 20 ng ml$^{-1}$ of bFGF

(Invitrogen, CA, USA), 50 ng ml$^{-1}$ human EGF (Invitrogen), N2 (Invitrogen), B27 (Invitrogen), 10 μM ROCK inhibitor (Enzo Life Sciences, NY, USA), and 1% penicillin/streptomycin (Gibco, OK, USA).

To evaluate tissue quality, 5 μl of the 100 μl cell suspension was put on a slide. After drying the medium in room temperature (RT), the cells were fixed in 100% ethanol followed by standard H&E staining. In parallel, 50 μl of the 100 μl cell suspension was seeded onto collagen-I-coated dishes (Corning, NY, USA) and cultured as 2D-LC cells in MBM for 5–7 days. One hundred microliters Matrigel (Corning) was added to the remaining 50 μl suspension for establishing LCOs, and the resulting cell suspension was allowed to solidify on pre-warmed six-well culture plates (Corning) at 37 °C for 10 min. After gelation, 3 ml MBM was added to the well. The medium was changed every 4 days, and the organoids were passaged after 1–3 weeks. For passaging, a solidified Matrigel drop containing the LCOs was harvested using cold DPBS and then centrifuged at 112 × *g* for 3 min at 4 °C. The pellets were washed with cold DPBS and centrifuged at 250 rcf for 15 min at 4 °C. The organoids were resuspended in 2 ml TrypLE Express (Invitrogen) and incubated for 10 min at 37 °C for dissociation. Afterwards, 10 ml DMEM/F12 containing 10% FBS was added and centrifuged at 112 × *g* for 3 min. The pellets were washed with DPBS and centrifuged at 112 × *g* for 3 min. The pellets were resuspended in MBM + Matrigel (1:3) and reseeded at 1:3–1:4 ratios to allow the formation of new organoids. For single cell analysis, dissociated cells from organoids were seeded as single cells with Matrigel in in-house developed micro-well device. Only the wells containing a single cell were monitored. To culture NBOs, normal bronchial samples were used and the same steps as described above for LCO cultures were followed, except that the NBOs were cultured in MBM + WNA (MBM supplemented with 30% Wnt 3A conditioned medium, 100 ng ml$^{-1}$ Noggin (Peprotech, NJ, USA), and 500 nM A83-01 (Peprotech).

Organoids successfully cultivated during passage 3 were regarded as success in organoid formation. Three-fourth were cryopreserved in three vials and one-fourth was cultivated additional one passage (passage 4) to expand organoids amount, and cryopreserved in six vials. More specifically, after the diameter of organoids reached up 100–150 μm, organoids (>passage 3) were banked. To stock organoids, organoids were harvested using cold DPBS and then centrifuged at 112 × *g* for 3 min at 4 °C. The pellets were washed with cold DPBS and centrifuged at 250 rcf for 15 min at 4 °C. Supernatant was removed and organoid pellet was resuspended in freezing media; Culture Media 7: ES grade FBS (Gibco) 2: DMSO (Sigma) 1. The stock vials were stored in gas nitrogen tank.

**Histology and imaging**. Tissues and organoids were fixed in 4% paraformaldehyde (PFA) followed by dehydration, paraffin embedding, sectioning, and standard H&E staining. For IHC staining, the samples were incubated with primary antibodies including anti-napsin A (1:200 dilutions; #NCL-L-Napsin A, Novocastra, IL, USA), anti-TTF-1 (1:200 dilutions; #NCL-L-TTF-1, Novocastra), anti-CK7 (1:400 dilutions; #M7018, Dako, CA, USA), anti-p63 (1:200 dilutions; #M731701-2, Dako), anti-cytokeratin 5/6 (CK5/6; 1:200 dilutions; # MA5-12429, Invitrogen), anti-CD133 (1:200 dilutions; #64326, Cell Signaling Technology, MA, USA), anti-PD-L1 (1:200 dilutions; #13684, Cell Signaling Technology), and anti-c-Met (1:400 dilutions; #257261, Dako). The sections were subsequently incubated with secondary antibodies (#AI-2000, #AI-1000, Vector laboratories, CA, USA) for 1:5000 dilutions and visualised using the ultraView Universal DAB Detection kit (Ventana Medical Systems, AZ, USA). Nuclei were counterstained with Harris haematoxylin. Images were acquired on a Leica Eclipse E600 microscope.

**Genomic analysis**. DNA was extracted from tumour tissues with matched normal tissues and matched LCOs using the DNeasy Blood & Tissue Kit (Qiagen, Germany) according to the manufacturer's protocol. To evaluate the somatic mutations in the 12 paired tumour tissues and LCO lines, targeted next-generation sequencing was performed using the MiSeq platform (Illumina, Inc., CA, USA) with OncoPanel_AMCv1 (OP_AMCv1) to capture the exons of 164 cancer-related genes plus partial introns from 39 genes that are frequently rearranged in cancers and to detect fusion genes. Genomic DNA (200 ng) was fragmented to an average of 250 bp by sonication (Covaris, Woburn, MA, USA) followed by size selection with Agencourt AMPureXP beads. Each library was generated with sample-specific barcodes 6 bp in size and quantified with PicoGreen. Eight libraries were combined to yield a total of 720 ng for hybrid capture with an Agilent Sure Select XT custom Kit (OP_AMCv1 RNA bait; Agilent Technologies). The concentration of the enriched target was measured by quantitative polymerase chain reaction (qPCR; Kapa Biosystems, Inc., Woburn, MA, USA) and loaded on the MiSeq platform (Illumina Inc., San Diego, CA) for 75 bp paired end sequencing. Whole exome sequencing was performed using the Illumina Hiseq 2500 platform in five cases on fresh early passage organoids and matched late passage organoids, and in seven cases on fresh organoids, matched formalin-fixed & paraffin-embedded (FFPE) primary tumour tissues and background non-neoplastic tissues. A SureSelect Exome Enrichment kit version 6.0 (Agilent Technologies, Santa Clara, CA, USA) was used to capture the whole exome regions for whole exome sequencing. The DNA libraries were sequenced for 150 bp-paired end sequencing. Illumina sequencing data were processed using the Genome Analysis Toolkit (GATK) v1.6.5.[64] Sequenced reads were aligned to the human reference genome (NCBI build 37) with the Burrows–Wheeler Aligner (version 0.5.9)[65] using the default options. De-multiplexing was performed using MarkDuplicates of the Picard

package to remove PCR duplicates. De-duplicated reads were realigned at known indel positions with the GATK IndelRealigner, and the base quality was recalibrated using the GATK Table Recalibration[64,66]. Somatic variant calling for single nucleotide variants and short indels was performed with matched normal tissues using MuTect (v1.1.7)[67] and SomaticIndelocator in GATK, respectively. Germline variants from somatic variant candidates were filtered out with common dbsnp (build 141; found in ≥1% of samples), the Korean Reference Genome Database (KRGDB), and a panel of normal samples. The final somatic variants were annotated using the Variant Effect Predictor (v79) and then converted to a mutation annotation file format using vcf2maf. Quality checks for fastq files were performed using FastQC [http://www.bioinformatics.babraham.ac.uk/projects/fastqc]. Quality checks for analysis-ready BAM files, including the total read number, % PF reads, % selected bases, mean target depth, % target not covered, % target bases covered 30×, and % duplication for cleaned BAM files were performed using the CalculateHSMetrics and MarkDuplicates modules of the Picard programme. Integrative Genomics Viewer (IGV) was used to view the BAM files[68]. R software (v3.0) with maftools[69] and ggplot2 packages was used for data visualisation. To detect CNAs, BAM files were analysed using CNVkit[68] for LCOs and normal tissues. Average mean target coverage of organoid sequencing was 176×. Fingerprinting analysis was performed using the panel detecting 24 SNP alleles on the Sequenom MassARRAY technology platform (Sequenom, CA, USA).

**Immunofluorescence**. For IF staining, LCOs were harvested, fixed in 4% PFA for 15 min, and dehydrated with 30% sucrose overnight at 4 °C. After sucrose removal, LCOs were placed in moulds containing optical coherence tomography (OCT) compound and solidified. The resulting LCO samples were sectioned, permeated with 0.5% TritonX-100/PBS for 5 min, and blocked with 3% bovine serum albumin/PBS. They were incubated with primary antibodies overnight at 4 °C. Primary antibodies were detected by incubating with Alexa Fluor 488- (1:1000 dilutions; #A-11029, Invitrogen) and 594-conjugated secondary antibodies (1:1000 dilutions; #A-11037, Invitrogen) for 1 h at RT. To delineate the cellular organisation of NBOs and LCOs, anti-pancytokeratin (Pan CK; 1:1000 dilutions; #4545, Cell Signaling Technology), anti-p63 (1:1000 dilutions; #4892, Cell Signaling Technology), anti-mucin 1 (MUC1; 1:500 dilutions; #ab45167, Abcam, MA, USA), anti-CC10 (1:200 dilutions; #sc-130411, Santa cruz, TX, USA), anti-keratin 5 (KRT5; 1:200 dilutions; #905504, BioLegend, CA, USA) and anti-keratin 7 (KRT7; 1:200 dilutions; #4465, Cell Signaling Technology) were used to stain lung epithelial cells, basal cells, goblet cells and club cells, respectively. Cilia were co-stained with anti-ARL13B (1:1000 dilutions; #17711-1-AP, Proteintech, IL, USA) and anti-acetylated α-tubulin (Ac-Tub; 1:1000 dilutions; T7451, Sigma, CA, USA). Nuclei were counterstained with DAPI (1:2000 dilutions; D9542, Sigma) for 15 min, and imaging was performed on a Carl Zeiss fluorescence microscope (Carl Zeiss, Germany).

**Human NSCLC xenograft tumour models**. The entire experimental protocol was conducted in compliance with the institutional guidelines and approved by the Institutional Animal Care and Use Committee (IACUC) of the Asan Institute for Life Sciences, Asan Medical Center, Korea. To generate xenograft tumour models, each cancer tissue was engrafted by three different methods: direct transplantation, injection of 2D cultured primary lung cancer cells (2D-LC cells) or dissociated LCO and 3D-transplantation of cultured LCO. Two-thirds of the surgically resected LC tissues from patients were directly engrafted into NOD *scid* gamma mice (NOD.*Cg-Prkdc^scid Il2rg^tm1Wjl*/SzJ; 6 weeks old). The remaining one-third was used equally to culture 2D-LC cells and LCOs by followed the methods as described above for 2D-LC and LCO cultures. After culturing 2D-LC cells, $2 \times 10^6$ 2D-LC cells were harvested for injection. One hundred microliters of the 2D-LC cells mixed in Matrigel were subcutaneously injected into NOD *scid* gamma mice. To engraft LCOs, LCOs cultured in 3–4-wells of 24-well plates were harvested and separated from Matrigel by centrifugation at 250 *rcf* for 15 min at 4 °C. To engraft the LCOs with 3D structures, the LCOs separated from Matrigel were resuspended in 30–50 μl new Matrigel. The Matrigel containing LCOs was solidified on mixed cellulose ester membranes (Millipore, Germany) and subcutaneously transplanted into NOD *scid* gamma mice (Supplementary Fig. 4). To inject LCOs that were dissociated as single cells, the LCOs separated from Matrigel were dissociated by resuspension in 2 ml TrypLE Express followed by incubation for 10 min at 37 °C. One hundred microliters of dissociated LCOs were mixed in Matrigel and subcutaneously injected into NOD *scid* gamma mice. Excepting for direct tissue transplantation, two different methods (i.e. transplantation of LCOs with 3D structures and injection of 2D-LC cell, transplantation of LCOs with 3D structures and injection of dissociated cells from LCOs) were used together into each left and right back side of a same mouse. We monitored the mice for 3 months after engrafting and recorded the date when tumours grew up over 100 mm³. Tumours that successfully formed in the mice were harvested and used for H&E and IHC staining. Tumour growth was measured using callipers. The estimated volume was calculated according to the formula: Tumour volume (mm³) = 0.5 × length ×width².

**Drug screening**. LCOs cultured in 24-well plates over 2 weeks were harvested and dissociated using TrypLE Express. The dissociated LCOs were mixed in MBM + Matrigel (1:3 ratio) and seeded onto 96-well white plates (10 μl of $2 \times 10^3$ cells per a well; Corning). After gelation, 100 μl MBM was added to each well. The LCOs were allowed to grow for 10–14 days. Then, 10 concentrations of docetaxel, olaparib, erlotinib, and crizotinib (all Selleckchem, TX, USA), and DMSO controls were added every 3 days in triplicate. After 6 days, the medium was changed to 100 μl MBM per well to measure cell viability, and 100 μl CellTiter-Glo (Promega) was added to each well. The plates were agitated for 30 min at RT prior to luminescence reading. The determination of IC₅₀ values was conducted using Graph Pad Prism.

**Drug sensitivity test in PDX models**. The entire experimental protocol was conducted in compliance with the institutional guidelines and approved by the institutional animal care and use committee (IACUC) of the Asan Institute for Life Sciences, Asan Medical Center, Korea. After transplanting cancer organoids into NOD *scid* gamma mouse, we monitored the mice for 3 months. When the tumour size reached over 100 mm³, we retransplanted tumours from NSG to nude mouse. After the nude mice had reached a volume of 80–120 mm³, animals were randomised (four mice with tumours on the flank per group) and administered drugs by intraperitoneal injection: Erlotinib (50MPK, seven times per a week), Crizotinib (50MPK, seven times per a week) and Olaparib (50MPK, seven times per a week). All drugs were dissolved in DMSO. Tumour dimensions were measure five times a week with callipers. Tumours were harvested for further analysis. Mice were sacrificed ~1 month after chemical injection or earlier if tumours reached a size >2000 mm³, body weight loss exceeded 20%. Final tumour volumes were compared using two-tailed ANOVA adjusted for multiple comparisons.

**Western blotting**. Harvested cultured LCOs were lysed in lysis buffer (Cell Signaling Technology, MA, USA) containing phosphatase inhibitor cocktail C (Santa Cruz). The concentration of proteins in cell lysates was quantified by the Enhanced BCA Protein Assay Kit (Pierce Biotechnology, Inc., MA, USA) and 30 μg of proteins were loaded in each lane. Proteins were subjected to SDS–PAGE, transferred to nitrocellulose membranes with a 0.45 μm pore size (Amersham, GE health care life sciences, PA, USA). The membranes were blocked for 1 h at RT with 5% skim milk (BD Difco, NJ, USA) in 1xTris-buffered saline Tween-20 (TBST) (25 mM Tris, 150 mM NaCl, 2 mM KCl, pH 7.4, supplemented with 0.1% Tween-20). Primary antibodies diluted with 5% skim milk in 1X TBST were incubated overnight at 4 °C: anti-phosphorylated Met (pMet; 1:1000 dilutions; #3135, Cell Signaling Technology), Met (1:1000 dilutions, #4560, Cell Signaling Technology), phosphorylated EGFR (pEGFR; 1:1000 dilutions, #2234, Cell Signaling Technology), EGFR (1:1000 dilutions, #2232, Cell Signaling Technology), phosphorylated Akt (pAkt; 1:1000 dilutions, #9271S, Cell Signaling Technology), Akt (1:1000 dilutions, #9272S, Cell Signaling Technology), phosphorylated Mek (pMek; 1:1000 dilutions, #9121S, Cell Signaling Technology), Mek (1:1000 dilutions, #9122S, Cell Signaling Technology), phosphorylated Erk 1/2 (pErk; 1:1000 dilutions, #9101S, Cell Signaling Technology), Erk 1/2 (Erk; 1:1000 dilutions, #SC-135900, Santa Cruz) and GAPDH (1:1000 dilutions; #SC-32233, Santa Cruz). After incubating primary antibody, the membranes were washed with 1X TBST three times for 30 min. Incubation with HRP-conjugated goat anti-rabbit (#ADI-SAB-300-J) or anti-mouse (#BML-SA204-0100) IgG secondary antibodies (1:5000; Enzo Life Sciences, Inc., NY, USA) antibodies was performed for 1 h at RT. The membranes were washed with 1X TBST three times for 30 min, and the antigen-antibody reaction was visualised with an enhanced chemiluminescence assay (Amersham, GE health care life sciences). The full blots of the cropped images are presented in Supplementary Fig. 9.

**Reporting summary**. Further information on research design is available in the Nature Research Reporting Summary linked to this article.

## Data availability
The authors declare that all data that support the findings of this study are available from the corresponding author upon request. The source data underlying Figs. 3, 6a–c, f and Supplementary Figs. 3, 8 are provided as a Source Data file. All sequencing data that support the findings of this study have been deposited in the European Genome-phenome Archive (EGA). The sequencing data for targeted or whole exome sequenced samples are available under study accession EGAS00001003786.

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

## Acknowledgements

This study was supported by the Technology Innovation Programme for Fostering New Post-Genome Industry funded by the Ministry of Trade, Industry & Energy (MOTIE, Korea) (#10067796 and #10067407), Basic Science Research Programme through the National Research Foundation of Korea (NRF), and the Ministry of Education (#2018042382), as well as a grant from the Asan Institute for Life Sciences, Asan Medical Center (Seoul, Korea) (#2018-491). The bio-specimens and data used in this study was provided by Asan Bio-Resource Center, Korea Biobank Network. We also thank Dr. Joon Seo Lim from the Scientific Publications Team at Asan Medical Center for his editorial assistance in preparing this manuscript.

## Author contributions

S.J.J. and M.K. conceived and designed the study; M.K., H.M., H.-J.J., D.J.J., T.H.S. and G.S.J. performed the experiments; M.K., H.M., E.K.C. and S.-Y.J. acquired in vivo data; D.K.K. resected and offered cancer tissues from lung cancer patients; C.O.S., E.J.C. and S.-M.C. analysed and studied the whole genomic data; S.J.J., M.K. and H.M. analysed the data; and S.J.J., M.K., C.O.S., A.M.T., S.J. and M.M. wrote the manuscript with contributions from all co-authors.

## Additional information

**Competing interests:** The authors declare no competing interests.

**Peer Review Information**: *Nature Communications* thanks Hans Clevers and other anonymous reviewers for their contribution to the peer review of this work. Peer reviewer reports are available.

