## [Peer Review File · Nature Communications]

Reviewers' comments:

Reviewer #1 (Expertise: Cancer organoid biobank, Remarks to the Author):

In their study Kim et al report the generation and characterization of 62 patient-derived lung cancer organoids and 5 normal bronchial organoids derived from normal tumour-adjacent tissue. The authors used these organoids to perform proof-of-principle drug screening experiments in which they show that lung cancer organoids can potentially be used to predict patient response.

This is a highly descriptive story. The characterization of their lung cancer organoids, however, is somewhat superficial. Indeed, their comparison of 2D cell line, PDX, and organoid generation from the same sample could be informative if it were accompanied with a more comprehensive characterization of these samples. The same is true of their drug sensitivity experiments.

Overall, the lung cancer organoid lines the authors have generated and partially characterized represent a useful tool that can be additive to the field of lung cancer research. The study should therefore be eventually published but not before some major revisions.

General:

1. All the images provided only show single organoids with what looks like debris surrounding it. The authors should provide a more comprehensive view of the well in which the tumour and normal organoids are growing to prove their point of optimal growth in all conditions. A stain for a proliferative marker like Ki67 would moreover prove the author's claim. This should be done for both tumour and normal organoids.

2. The authors do not show that their organoid lines are genetically stable across multiple passages. Moreover, they do not show that the cell-type composition of their organoid lines is stable across multiple passages. Genetic stability could be shown by CNV analysis or karyotyping or cells from early and late passages. The comparison between early and late passage is essential to show the value of the system.

3. In the field of lung cancer, multiple cell lines are available. The authors do not show the comparison between the efficiency of growing organoids and 2d cell lines. Proving the higher efficiency of organoids formation over 2d cell lines will boost the impact of the method.

4. Figure 1:

- a. In figure 1b, the images of day 30 in all passages are suboptimal and make it difficult to determine the structure of the organoids.
- b. Figure 1d would be aided by a diagram showing the prevalence of the tumour types in patients. This would show whether or to what degree their biobank is representative.

5. Figure 2:

- a. In figure 2a the authors should point out the morphological or histological features of each tumour type that they reference in the text. Additionally, the authors could leave out 2a since HE-staining is already shown per tumour type in 2b-e. In this case, the authors should also provide detailed IHC analysis a representative large cell neuroendocrine carcinoma.
- b. The staining for CK7 in figure 2b appears overexposed. A detailed inset could show specificity of the staining.
- c. Figure 2d shows the heterogeneity which is maintained in culture but it is unclear whether the image is of one organoid with mixed phenotypes or two organoids with homogenous phenotype per organoid. In addition to that, it would be interesting to show the origin of the heterogenous populations by culturing single CK7+ or CK5/6+ cells. Would these only grow out to be the phenotype of their origin or do they give rise to a heterogenous population?

d. The staining for CD56 in In figure 2e is not convincing. Other markers like Chromogranin A or synaptophysin would strengthen their claim that this organoid maintains the neuroendocrine features of the corresponding primary tumour tissue.

6. In addition to the oncopanel of early passage organoids, a whole genome or exome sequencing would strengthen the comparison between organoids and tumour. The correlation of MAF does not seem to be a strong correlation. Many outliers lie in the maximum range of organoid MAFs.

7. Figure 3:

a. Although the authors reference the generation of iPSC derived normal bronchiolar organoids, the normal bronchiolar organoids they describe are more similar to adult stem cell derived organoids, Sachs et al. BioRxiv (2018). The morphology of these organoids is strikingly different to what the authors show. Moreover, the growth of their organoids looks suboptimal and late passage (P10) is missing from figure 3a.

b. In addition to ciliated, basal and goblet cells, the bronchus comprises mostly of club cells. The authors so not show the presence of club cells in their organoids.

c. Figures 3c-e: It is not clear why the authors show both day 7 and day 14 timepoints for their immunofluorescence. The images they present add confusion to their claims. In particular, the day 7 images in 2e show p63 negative cells which at day 14 are all p63 positive. The transition from p63+ tumour cells to negative cells and back to positive, remains unexplained and is not even addressed in the text. The cilia inserts should also be at higher magnification to prove the point of the authors of different phenotypes in the organoids.

8. Figure 4:

a. The authors do not mention the number of cells transplanted and whether the whole organoid transplantations were performed with the same number of cells or with a number of organoids that is comparable. This could influence to the success rate of engraftment.

b. In addition to ciliated, basal and goblet cells, the bronchus comprises mostly of club cells. The authors so not show the presence of club cells in their organoids.

c. Figure 4d and the results obtained for PD-L1 expression seem irrelevant to the story.

d. In figure 4c, LCO-43 is the only organoid line, which engrafts better after dissociated as opposed to as whole organoids. The authors should comment on this finding.

e. Although the authors reference the normal bronchiolar organoids, the deprivation of these organoids in more similar to adult stem cell derived organoids, which are developed by Sachs et al. BioRxiv (2018). The morphology of these organoids are strikingly difference to those shown by the authors. This difference should be addressed by the authors.

9. Figure 4:

a. The authors do not mention the number of cells transplanted and whether the whole organoid transplantation were performed with the same number of cells. This could be of major influence to the success rate of engraftment.

b. Figure 4d and the results obtained of PD-L1 seems irrelevant to the story.

c. In figure 4c, LCO-43 is the only organoid line, which engrafts better dissociated then as 3D. The authors should comment on this finding and explain potential mechanisms.

10. Figure 5:

a. All bar graphs of the IC50 values do not include the normal organoid line. This should be added to improve the findings. Additional normal lines should also be included in the drug assays. Also, they authors never state whether or not their 5 normal lines are matched to some of their tumour organoid lines. If they have matched normal lines for the tumour lines they use for the drug testing, this would improve the strength of the sensitivity difference.

b. In figure 5c, LCO-51 is used as EGFR missense organoid line, however, the line is not shown in figure 2f. This line is not included in the description of lines subjected to genomic analysis. Did they specifically check for this mutation in this line? Why did they choose this line?

c. Figure 5b, LCO-49 is more sensitive than the mutation carrying LCO-28. No comments were made by the authors.

11. Textual:

- a. Line 176 references figure 2f but is shown in 2g.
- b. Line 222-223 "In parallel using our new method, of lung cancer organoids were transplanted whole into the same mouse." It is not clear what they authors are trying to say.

Reviewer #2 (Expertise: Cancer genomics, Remarks to the Author):

In this manuscript by M Kim et al, they describe the development and application of in vitro organoid models of lung cancer (LC). In the first instance they describe methods for the derivation of organoids across different sub-types of LC and compare this to PDX models and 2D cell lines. They go onto examine how well organoids reflect the somatic mutation frequency and histology of primary tumours. Furthermore, the authors demonstrate that LC organoids are suitable for drug testing, including specific examples of differential drug response in defined molecular contexts, such as BRCA2 and EGFR mutations.

Although this manuscript adds to a long list of papers describing the derivation of tumour organoids from different cancers/histology, to the best of my knowledge this is the first description of tumour derived organoids from lung cancer. This is a valuable contribution, especially given the health burden of LC, and I expect the protocols developed will be of utility to many research labs.

Overall, I enjoyed reading the manuscript and felt that the study was well performed and the work is clearly presented. Nonetheless, I have a number of suggestions on how the manuscript should be improved prior to publication.

1. I believe the most valuable aspect of this work is the description of the derivation method and consequently this should be extremely well documented. Although the protocols are well described, some key pieces of information are absent or not clearly reported:
 - a. High success rates for derivation have been reported in the literature for organoids from other tissues, which have not always been reproduced in other labs. This can be misleading and it can be difficult to understand the underlying reasons for these differences. For this study how was derivation success measured? What are the criteria for a successful LC organoid derivation? Does this include the ability to cryopreserve organoids and reconstitute them from frozen? Is there a minimum number of passages or size of cell bank that needs to be achieved? These points (and any other relevant ones) should be clearly articulated in the manuscript.
 - b. The authors use a pre-screening tissue evaluation protocol prior to derivation. This is a sensible approach but success rates should be clearly reported with and without this pre-screening step. For example, the authors report an 87% success rate after screening, but this would actually be 55% if all samples were included.
 - c. Was there any bias in the ability to derive organoids from different molecular sub-types or histology? This is not clearly presented in the text.
 - d. What are the success rates for derivation of normal bronchial organoids?
2. The comparison of organoids with 2D cell lines and PDX is an interesting aspect of this study and could have implications on the usage of different types of models. Were there any specific LC sub-types/histology where you could derive a PDX and not an organoid, or vice-versa? What was the derivation success rate for 2D cell lines and again was there a bias?
3. The genetic analysis of the organoids is limited to targeted gene sequencing in only 12 models. This is considerably more limited in scope than the in-depth analyses of organoids cultures which have been performed in other cancer types. Sequencing a larger number of models would help to better understand if the models truly reflect the spectrum of patients in the clinic, and what molecular sub-types are represented by the organoids, and if there is any selection in the set of models which have been derived? Similarly, do the PDX, 2D cell lines and organoids represent different molecular sub-types? Indeed, the high frequency of BRCA2 mutations in this cohort is surprising for LC (20% of sequenced models), and could suggest some bias in the collection of organoids. At the moment, the analysis is underpowered to address these points convincingly.

4. The drug testing results are interesting aspect of this study.
 - a. The generation of patient-matched organoids and PDX models, and the ability to compare these, is a strength of this manuscript and therefore it is surprising that drug response has not been compared for select examples. For example, is the differential sensitivity of LCO-28 and LCO-55 to Olaparib preserved in both model systems? Table 1 indicates that organoids and PDX are available. I would usually hesitate to ask for such an experiment, but the authors clearly demonstrate they have this capability and this would be of significant interest to the community to compare the response in organoids and patient matched PDX models.
 - b. How many replicates were measured for the dose response curves in Figure 5 a,b and c?
 - c. The dose response curves would be easier to interpret if the data was plotted in μM but on a log scale (rather than log transformed values).
 - d. Why are there multiple values for MET copy number for each organoid in panel 5d?
5. The authors indicate that LC organoids could be used for personalized care, so called patient 'avatars'. This may be true in the future but this has not been demonstrated in this manuscript and a more nuanced presentation of their work would be more appropriate. Specific points include:
 - a. The Title of the manuscript should be altered because the authors do not demonstrate that these are personalised models that could inform the care of an individual patient.
 - b. What is the timeline to generate organoids suitable for drug testing? How would this fit with the clinical path for a patient?
 - c. How do the authors believe a personalised LC organoid approach would be used in the clinic? Many LC patients are cured surgically and therefore do the authors imagine this would be implemented in the setting of recurrent or metastatic disease? If so, are the authors able to derive LC organoids from fine needle biopsy samples?

Reviewer #3 (Expertise: Lung cancer, Remarks to the Author):

In their manuscript, Kim and colleagues describe an approach for derivation of patient-specific lung cancer organoids that appears broadly suitable for culturing epithelial cells from 5 major cancer sub-types. They demonstrate the potential of these organoids for testing targeted molecular therapies and show that in the majority of cases the mutational spectrum of the cells is maintained *ex vivo*. They also show an enhanced capacity and rapidity of these cultured organoids to initiate tumors in subcutaneous xenotransplantation assays compared with conventional (non-3D) approaches. The work is of high quality and the findings are credible, but there is no new biology revealed, so the manuscript seems better suited as a technical report for lung cancer researchers rather than a research article.

Major criticism:

Because sequencing-based approaches can already identify the appropriate targeted therapy for the oncogenic mutations in any given patient, culture-based assays are currently only potentially useful for research applications. In this regard, there is no evidence provided that the ability to grow cancers as organoids provides any additional value over routine 2D culture, beyond establishing PDX models.

Minor criticism:

Mutations in one of the organoids (LC-30) are reported to have zero concordance with mutations in the parental cancer. What do the authors make of this result? Does it suggest that the tumor cells did not grow and that the organoid was derived entirely of non-malignant lung epithelial cells admixed with the tumor? In any case, this result is concerning that the application of this organoid assay for patient-specific drug testing (or other applications) may be risky without first sequencing every organoid to confirm it is comprised of malignant cells.

Response to referees letter

We greatly appreciate the reviewers' thoughtful and insightful suggestions. We are sure that additional experiments and extensive revisions as suggested by reviewers have significantly improved quality of our manuscript. All expressed concerns have been addressed with necessary changes made to the manuscript and with additional analyses where appropriate. Our specific responses are following:

Specific responses to reviewers

Reviewer #1 (Expertise: Cancer organoid biobank, Remarks to the Author):

In their study Kim et al report the generation and characterization of 62 patient-derived lung cancer organoids and 5 normal bronchial organoids derived from normal tumour-adjacent tissue. The authors used these organoids to perform proof-of-principle drug screening experiments in which they show that lung cancer organoids can potentially be used to predict patient response.

This is a highly descriptive story. The characterization of their lung cancer organoids, however, is somewhat superficial. Indeed, their comparison of 2D cell line, PDX, and organoid generation from the same sample could be informative if it were accompanied with a more comprehensive characterization of these samples. The same is true of their drug sensitivity experiments.

Overall, the lung cancer organoid lines the authors have generated and partially characterized represent a useful tool that can be additive to the field of lung cancer research. The study should therefore be eventually published but not before some major revisions.

Response: *We agree with reviewer's opinion that more comprehensive characterization of the organoids and drug sensitivity screening will strengthen our study. We added more organoid lines and performed experiments for additional characterization of organoids, freeze-thaw experiments, single cell or single organoid analysis, additional genomic study and drug tests using PDX in a limited time and budget, and the results were included in **the main text or supplementary data of the revised manuscript.***

General comments:

Comment 1: All the images provided only show single organoids with what looks like debris surrounding it. The authors should provide a more comprehensive view of the well in which the tumour and normal organoids are growing to prove their point of optimal growth in all conditions. A stain for a proliferative marker like Ki67 would moreover prove the author's claim. This should be done for both tumour and normal organoids.

Response 1: As reviewer mentioned, we changed the images of both tumour and normal organoids in **Figure 1b and Figure 4a in the revised manuscript** and added a more comprehensive view of the wells in **Supplementary Figure 1a-c of the revised manuscript**. Also, we stained organoids with a proliferative marker Ki67 and this data was displayed in **Supplementary Figure 1a-c of the revised manuscript**. As shown in the data substantial parts of the tumour organoids and normal organoids expressed Ki67. We included the explanation **on p.5 lines 13-19 in the Results section of the revised manuscript**.

Comment 2: The authors do not show that their organoid lines are genetically stable across multiple passages. Moreover, they do not show that the cell-type composition of their organoid lines is stable across multiple passages. Genetic stability could be shown by CNV analysis or karyotyping or cells from early and late passages. The comparison between early and late passage is essential to show the value of the system.

Response 2: We agree with reviewer's opinion. To investigate genetic stability, we performed whole exome sequencing (WES) with CNV analysis using 7 sets of primary lung cancer tissue, normal lung tissues and matched organoids, and 5 pairs of early passage organoids and corresponding late passage organoids in a given limited time and budget. Major cancer driver genes, such as TP53 and EGFR were retained in all organoids (**Figure 3c of the revised manuscript**). In detected all mutations, most variants in primary tumour tissues were also retained in their corresponding organoids (**Figure 3c of the revised manuscript**). Comparing primary tumour tissue and organoid, major cancer driver genes, such as TP53 and EGFR were retained in late passage organoids (**Figure 3e of the revised manuscript**). In copy number analysis, most copy number alterations were preserved in late passage organoid and overall patterns of copy number variations were similar between early passage organoid and late passage organoid (**Supplementary Figure 3d of the revised manuscript**).

Comment 3: In the field of lung cancer, multiple cell lines are available. The authors do not show the comparison between the efficiency of growing organoids and 2d cell lines. Proving the higher efficiency of organoids formation over 2d cell lines will boost the impact of the method.

Response 3: As reviewer's recommendation, we added data of the comparison between the efficiency of growing organoids and 2D cell lines mentioned. In our experiments, efficiencies of monolayer culture and organoid formation from the same lung cancer patient tissue were similar. As shown in **Figure 1d of the revised manuscript and Table 1 of the original manuscript**, a primary lung cancer cells were established from 23 of 36 samples and lung cancer organoids were formed from 20 of 36 samples. To make this part more specific, we added the explanation **on p.5 line 30 and p.6 lines 1-10 in the Results section of the revised manuscript**. Notably, 2D cultured primary lung cancer cells could not be cultured for long term in

this 2D culture condition. After passage 5, their growth rate slowed and cells underwent senescence (Data not shown). This limited availability of 2d cultured primary cancer cell has been shown in other published papers^{1, 2}. Under this consideration, lung cancer organoid models have greater long term potential than 2d cultured primary lung cancer cells.

Comment 4 (Figure 1)

a. In figure 1b, the images of day 30 in all passages are suboptimal and make it difficult to determine the structure of the organoids.

Response 4a: *We changed the all images in **Figure 1b** of the original manuscript with better quality.*

b. Figure 1d would be aided by a diagram showing the prevalence of the tumour types in patients. This would show whether or to what degree their biobank is representative.

Response 4b: *In general population, the incidence of lung cancer subtypes other than 5 major subtypes included in our organoid biobank is less than 5 %³. To show success rates of organoid formation of various lung cancer subtypes more clearly, we added diagrams showing number of initial cytologic quality check (Cytologic QC), success rates of 2D culture, organoid culture and patients' derived xenograft (PDX) with detailed subtype information in **Figure 1d of the revised manuscript**. Since our organoid biobank project is ongoing, we have further increased the number of lung cancer organoid samples stored in organoid biobank to 80 lines including pre-existing 62 lines by following our protocol. So, we changed a diagram showing the tumour types of lung cancer organoids stored in the biobank in **Figure 1e of the revised manuscript**.*

Comment 5 (Figure 2)

a. In figure 2a the authors should point out the morphological or histological features of each tumour type that they reference in the text. Additionally, the authors could leave out 2a since HE-staining is already shown per tumour type in 2b-e. In this case, the authors should also provide detailed IHC analysis a representative large cell neuroendocrine carcinoma.

Response 5a: *We deleted H&E staining images in **Figure 2a of the original manuscript** and added IHC analysis of a large cell neuroendocrine carcinoma organoid and tissue in **Figure 2d of the revised manuscript**.*

b. The staining for CK7 in figure 2b appears overexposed. A detailed inset could show specificity of the staining.

Response 5b: As reviewer mentioned, we changed the CK7 staining image. However, CK7 is usually strongly expressed in pulmonary adenocarcinoma⁴.

c. Figure 2d shows the heterogeneity which is maintained in culture but it is unclear whether the image is of one organoid with mixed phenotypes or two organoids with homogenous phenotype per organoid. In addition to that, it would be interesting to show the origin of the heterogenous populations by culturing single CK7+ or CK5/6+ cells. Would these only grow out to be the phenotype of their origin or do they give rise to a heterogenous population?

Response 5c: In single organoid analysis, adenocarcinoma organoids were heterogeneous. Some organoids were a mixture of p63+ cells and p63- cells. Some organoids were CK7+, other organoids were CK5/6+ and the other organoids were a mixture of CK7+ and CK5/6+ cell. In single cell analysis, only few single cells grow to form organoid or organoid-like structure and most wells seeded by a single cell failed to grow to an organoid. Within the organoid, tumour-initiating single cells and terminally differentiated single cells may have differential abilities to form an organoid in our organoid culture conditions. Using immunofluorescence, we showed that the organoid like structure formed from a single cell seems to be composed of both adenocarcinoma and squamous carcinoma components. So we added this explanation on p.6, lines 28 and p.7, line 1-13 in the **Results of the revised manuscript** and in **Figure 2f of the revised manuscript**. Interestingly, wells containing two or three cells generated an organoid having a round shape. However, wells containing a single cell did not maintain a round shape organoid. However, we have not fully elucidated this result yet but plan to pursue further studies.

d. The staining for CD56 in figure 2e is not convincing. Other markers like Chromogranin A or synaptophysin would strengthen their claim that this organoid maintains the neuroendocrine features of the corresponding primary tumour tissue.

Response 5d: As reviewer's suggestion, we stained another neuroendocrine marker synaptophysin to small cell carcinoma tissue and organoids. The small cell carcinoma organoids expressed the synaptophysin as well as CD56 as in the corresponding tumour tissue. We displayed this result in **Figure 2e of the revised manuscript**.

Comment 6: In addition to the oncopanel of early passage organoids, a whole genome or exome sequencing would strengthen the comparison between organoids and tumour. The correlation of MAF does not seem to be a strong correlation. Many outliers lie in the maximum range of organoid MAFs.

Response 6: We appreciate the point of the reviewer that the MAF correlation is not as strong as it should be. However, the reason of having many outliers at the maximum range in the organoid MAFs is due to

the nature of our organoid culture, where lung cancer organoids grow without any stromal or immune cell contamination. Therefore, if a mutation is common in all cancer cells, it will be represented in all sequencing data and thereby reaching to the highest score (=1). Nevertheless, the overall trend clearly results in a good positive correlation. Our oncopanel also confirms the paired relationship between tumour and organoid samples. In the revised manuscript, we made this point clearer (p.8, line 5-16) so that readers can appreciate the meaning of our data, following our original intention. In whole exome analysis, major cancer driver genes such as TP53 and EGFR were retained in all organoids (Figure 3c of the revised manuscript). In detected mutations, most variants in primary tumour tissue were also retained in organoids (Figure 3c of the revised manuscript), and the distribution of VAF was also well maintained in organoid sample likely primary tumour sample (Figure 3d of the revised manuscript). In the comparison between primary tumour tissue and organoid, major cancer driver genes such as TP53 and EGFR were retained in the late organoid (Figure 3e of the revised manuscript) and the distribution of VAF was also retained in both early passage and late passage organoid (Figure 3f of the revised manuscript).

Comment 7 (Figure 3):

a. Although the author's reference the generation of iPSC derived normal bronchiolar organoids, the normal bronchiolar organoids they describe are more similar to adult stem cell derived organoids, Sachs et al. BioRxiv (2018). The morphology of these organoids is strikingly different to what the authors show. Moreover, the growth of their organoids looks suboptimal and late passage (P10) is missing from figure 3a.

Response 7a: We added images of normal bronchial organoids in late passage (P12) in Figure 4a of the revised manuscript and the explanation on p.9 line 21-24 in the results section of the revised manuscript. Sachs et al. BioRxiv (2018) showed normal bronchiolar organoids i.e. representing peripheral conducting airway. Our normal bronchial organoids are derived from large airway, and therefore, we used the term "bronchial organoid" rather than "bronchiolar organoid". Cellular composition of peripheral conducting airway and large airway is different. Moreover, the medium component in Sachs et al. BioRxiv (2018) was different with our medium. The morphological difference of the bronchial organoids may also be influenced by difference of culture media. However, the cellular composition and spatial patterning of cell types in our NBOs recapitulates airway tissue architecture. As with the Sachs et al group, our organoids are composed of an outer layer of basal cells with an inner luminal lining of differentiated ciliated and secretory cells, which we show through immunofluorescence.

b. In addition to ciliated, basal and goblet cells, the bronchus comprises mostly of club cells. The authors so not show the presence of club cells in their organoids.

Response 7b: As reviewer mentioned, we investigated the presence of club cells in our bronchial organoids by staining with CC10. Club cells which are non-ciliated, non-mucous and secretory cells are usually found in peripheral conducting airway such as terminal bronchiole and respiratory bronchiole⁵. So we predicted that the cells consisting our normal organoids rarely include club cells. As shown in **Figure 4c of the revised manuscript**, CC10 positive cells were rarely found in some organoids. In the revised manuscript, we described clearly (**p.9, line 25-30, and p.13, line 28-30 and page 14, line 1-7 in the Results of the revised manuscript**).

c. Figures 3c-e: It is not clear why the authors show both day 7 and day 14 timepoints for their immunofluorescence. The images they present add confusion to their claims. In particular, the day 7 images in 2e show p63 negative cells which at day 14 are all p63 positive. The transition from p63+ tumour cells to negative cells and back to positive, remains unexplained and is not even addressed in the text. The cilia inserts should also be at higher magnification to prove the point of the authors of different phenotypes in the organoids.

Response 7c: We fully agreed with the reviewer's comments. To avoid confusion, we removed data of the day 7 images.

Comment 8 (Figure 4):

a. The authors do not mention the number of cells transplanted and whether the whole organoid transplantations were performed with the same number of cells or with a number of organoids that is comparable. This could influence to the success rate of engraftment.

Response 8a: For xenograft experiments, we cultured organoids using 24 well plates. Organoids of 4 wells were dissociated, and cells and organoids of another 4 wells were maintained whole organoids having 3D structures, and transplanted into a mouse. Therefore the numbers of cells were approximately the same. To avoid the confusion, we changed the explanation as follows; "Consistent with previously described methods, we dissociated lung cancer organoids cultured in 4 of 24 wells into single cells and subcutaneously injected them into immunodeficient mice. In parallel using our new method, the same number of lung cancer organoids were transplanted whole into the same mice." **on p.10 line 21-29 in the Results of the revised manuscript**.

b. In addition to ciliated, basal and goblet cells, the bronchus comprises mostly of club cells. The authors so not show the presence of club cells in their organoids.

Response 8b: We added the data and explained it **in the Results of the revised manuscript**.

c. Figure 4d and the results obtained for PD-L1 expression seem irrelevant to the story.

Response 8c: *We moved these results to Supplemental Figure 7 of the revised manuscript.*

d. In figure 4c, LCO-43 is the only organoid line, which engrafts better after dissociated as opposed to as whole organoids. The authors should comment on this finding.

Response 8d: *We modified X-scale of “Day” in Figure 4c in the original manuscript to “Time to detect tumour” in Figure 5c of the revised manuscript, and added more detailed description in the main text of the revised manuscript.*

e. Although the authors reference the normal bronchiolar organoids, the deprivation of these organoids in more similar to adult stem cell derived organoids, which are developed by Sachs et al. BioRxiv (2018). The morphology of these organoids are strikingly difference to those shown by the authors. This difference should be addressed by the authors.

Response 8e: *We think that the morphology of organoids is closely associated with the region of resected tissues used for organoid culture and medium components such as growth factors and small molecules. So we added the explanation as follows; “The previous report showed normal lung organoids having round shapes, which were derived from peripheral conducting airway. However, our normal lung organoids had budding shapes not only round shapes, as cultured over passage 4. We suspect that morphological differences are associated with the tissue origin of anatomical sites and the culture media components such as growth factors and small molecules.” on p.13 line 28-30 and p.14 line 1-7 in the Discussion of the revised manuscript.*

Comment 9 (Figure 5):

a. All bar graphs of the IC50 values do not include the normal organoid line. This should be added to improve the findings. Additional normal lines should also be included in the drug assays. Also, they author never state whether or not their 5 normal lines are matched to some of their tumour organoid lines. If they have matched normal lines for the tumour lines they use for the drug testing, this would improve the strength of the sensitivity difference.

Response 9a: *We added IC50 values of the normal organoid line in Figure 6b, c and f of the revised manuscript. Unfortunately, we do not have matched normal organoid line for LCO-28, 43, 49, 51 and 55. We selected organoids for drug tests based on mutational profile by Oncopanel targeted NGS analysis. Drug assays for additional normal lines are possible, but we wonder if it is not potentiate our data.*

b. In figure 5c, LCO-51 is used as EGFR missense organoid line, however, the line is not shown in figure 2f. This line is not included in the description of lines subjected to genomic analysis. Did they specifically check for this mutation in this line? Why did they choose this line?

Response 9b: *The patient received PNA clamp based EGFR mutation analysis in his clinical course and the result showed same L858R mutation. Unfortunately, during the process tissue sample was lost and we could not perform Oncopanel analysis for the tissue sample.*

c. Figure 5b, LCO-49 is more sensitive than the mutation carrying LCO-28. No comments were made by the authors.

Response 9c: *We think the different sensitivity may not be associated with BRCA2 alteration because BRCA2 M956I alteration is predicted as non-damaging alteration. Another possible cause of sensitivity to Orlaparib is other cause of homologous recombination deficiency in LCO19.*

Comment 10 (Textual):

a. Line 176 references figure 2f but is shown in 2g.

Response 10a: *As reviewer mentioned, we changed it **in the revised manuscript**.*

b. Line 222-223 “In parallel using our new method, of lung cancer organoids were transplanted whole into the same mouse.” It is not clear what they author are trying to say.

Response 10b: *As reviewer mentioned, we changed it as follows; “In parallel using our new method, the same number of lung cancer organoids were transplanted whole into the same mice.” **on p.10 line 21-22 in the Results of the revised manuscript**.*

Reviewer #2 (Expertise: Cancer genomics, Remarks to the Author):

In this manuscript by M Kim et al, they describe the development and application of in vitro organoid models of lung cancer (LC). In the first instance they describe methods for the derivation of organoids across different sub-types of LC and compare this to PDX models and 2D cell lines. They go on to examine how well organoids reflect the somatic mutation frequency and histology of primary tumours. Furthermore, the authors demonstrate that LC organoids are suitable for drug testing, including specific examples of differential drug response in defined molecular contexts, such as BRCA2 and EGFR mutations.

Although this manuscript adds to a long list of papers describing the derivation of tumour organoids from different cancers/histology, to the best of my knowledge this is the first description of tumour derived organoids from lung cancer. This is a valuable contribution, especially given the health burden of LC, and I expect the protocols developed will be of utility to many research labs.

Overall, I enjoyed reading the manuscript and felt that the study was well performed and the work is clearly presented. Nonetheless, I have a number of suggestions on how the manuscript should be improved prior to publication.

Comment 1: I believe the most valuable aspect of this work is the description of the derivation method and consequently this should be extremely well documented. Although the protocols are well described, some key pieces of information are absent or not clearly reported:

a. High success rates for derivation have been reported in the literature for organoids from other tissues, which have not always been reproduced in other labs. This can be misleading and it can be difficult to understand the underlying reasons for these differences. For this study how was derivation success measured? What are the criteria for a successful LC organoid derivation? Does this include the ability to cryopreserve organoids and reconstitute them from frozen? Is there a minimum number of passages or size of cell bank that needs to be achieved? These points (and any other relevant ones) should be clearly articulated in the manuscript.

Response 1a: *We appreciate reviewer's points. In our lung cancer organoid biobank protocol, successful culture during passage 3 was considered as success and biobanked 3 vials, and additional one passage (passage 4) cultured to expand organoids amount and biobanked usually 6 vials. We added these protocol in **Materials & Methods section of the revised manuscript.***

*We performed re-thawing experiments to show whether cryopreserve organoids maintained the ability to reconstitute them from frozen. The results were shown in **Figure 1f and Table 3 of the revised manuscript** and described **in the results section on p.6 line 11 – 14 of the revised manuscript**.*

*So we changed a diagram showing the tumour types of lung cancer organoids stored in the biobank in **Figure 1e of the revised manuscript**. All of banked organoids were confirmed that they maintained the morphological features of original tissues.*

b. The authors use a pre-screening tissue evaluation protocol prior to derivation. This is a sensible approach but success rates should be clearly reported with and without this pre-screening step. For example, the authors report an 87% success rate after screening, but this would actually be 55% if all samples were included.

Response 1b: *We appreciate reviewer's points. For initial protocol development, we did trials and errors. After setting the protocol, we started with 36 samples for initial cytologic quality check and 23(64 %) samples were passed our criteria, and used for organoid culture and xenograft, and succeeded in 20 (87 %) and 13 (57 %) cases, respectively. Therefore, the success rate of generating lung cancer organoids and PDX from total starting 36 samples were about 56 % and 36 %, respectively. However, it has been already reported that successful in vitro cultivation using lung cancer cells from the tissues is dependent on the population of stromal elements such as fibroblast contained in lung cancer tissues⁶. So we think that the success rate of organoid formation purely related to culture protocol can be considered as about 87 %. According to our protocol, we further increased the number of lung cancer organoid samples stored in lung cancer organoid biobank to 80 lung cancer organoids including the existing 62 lung cancer organoids by following our protocol. To show this part more clearly, we added a figure showing overall feature of the experimental process from starting clinical samples to cytologic quality check (cytologic QC), 2D culture, organoid culture and PDX according to tumour subtypes in **Figure 1d of the revised manuscript**.*

c. Was there any bias in the ability to derive organoids from different molecular sub-types or histology? This is not clearly presented in the text.

Response 1c: *To answer reviewer's question clearly we added a figure showing overall feature of the experimental process from starting clinical samples to cytologic quality check (cytologic QC), 2D culture, organoid organoid culture and PDX according to tumour subtypes in **Figure 1d and Table 1 of the revised manuscript**. We could not find any differences in the ability to derive organoids according to different histologic subtypes. We think the differences according to molecular subtypes is extremely*

interesting. However, it was not able to check at this time because only successful cases were molecularly analysed.

d. What are the success rates for derivation of normal bronchial organoids?

Response 1d: *We cultured only 5 normal bronchial organoid using complete media, and all were successful, and then we stopped culturing normal organoids. So, we do not have enough data to mention the success rates.*

2. The comparison of organoids with 2D cell lines and PDX is an interesting aspect of this study and could have implications on the usage of different types of models. Were there any specific LC sub-types/histology where you could derive a PDX and not an organoid, or vice-versa? What was the derivation success rate for 2D cell lines and again was there a bias?

Response 2: *We established three types of lung cancer models, 2D cell line, PDX and organoid, by using 23 samples. 2D cell lines were established from 23 of 36 samples and lung cancer organoids were formed from 20 of 36 samples. But PDXs were generated from 10 of 36 samples. As shown in **Figure 1d of the revised manuscript and Table 1 of the original manuscript**, adenocarcinoma PDXs (4 of 13 samples, 31 %) showed lower success rate than squamous cell carcinoma PDXs (5 of 7 samples, 71 %). However, as the number of cases was small, we could not convince that a certain subtype of lung cancer has any selective advantage.*

3. The genetic analysis of the organoids is limited to targeted gene sequencing in only 12 models. This is considerably more limited in scope than the in-depth analyses of organoids cultures which have been performed in other cancer types. Sequencing a larger number of models would help to better understand if the models truly reflect the spectrum of patients in the clinic, and what molecular sub-types are represented by the organoids, and if there is any selection in the set of models which have been derived? Similarly, do the PDX, 2D cell lines and organoids represent different molecular sub-types? Indeed, the high frequency of BRCA2 mutations in this cohort is surprising for LC (20% of sequenced models), and could suggest some bias in the collection of organoids. At the moment, the analysis is underpowered to address these points convincingly.

Response 3: *We totally agree with reviewer's opinion. Given limited time and budget, we performed whole exome sequencing in 7 sets of primary lung cancer tissue, normal lung tissues and matched organoids, and 5 pairs of early passage organoids and corresponding late passage organoids. The results were shown in **Figure 3c-f of the revised manuscript and described in the results section on p.7 line 5 – 27 of the revised***

manuscript. We think the high frequency of BRCA2 mutations in our cohort is an incidental finding, because additional genomic analysis of our organoid lines did not show BRCA2 mutation.

4. The drug testing results are interesting aspect of this study.

a. The generation of patient-matched organoids and PDX models, and the ability to compare these, is a strength of this manuscript and therefore it is surprising that drug response has not been compared for select examples. For example, is the differential sensitivity of LCO-28 and LCO-55 to Olaparib preserved in both model systems? Table 1 indicates that organoids and PDX are available. I would usually hesitate to ask for such an experiment, but the authors clearly demonstrate they have this capability and this would be of significant interest to the community to compare the response in organoids and patient matched PDX models.

Response 4a: *As reviewer mentioned, we compared drug responses between lung cancer organoids and corresponding PDX models. The PDX corresponding to LCO-55, which has BRCA2 damaging alteration, showed sensitivity to olaparib. Also, the PDX corresponding LCO-51, which has EGFR L858R mutation and MET amplification, showed sensitivity only to crizotinib, but not to erlotinib. These data support that the drug sensitivity of lung cancer organoids is comparable with PDX models. This data was included in Supplementary Figure 8 of the revised manuscript. Unfortunately, we could not perform drug screening tests to PDX of LCO-28 and LCO-42 because we failed to generate enough number of PDXs to the drug screening tests. Even though we could not show complete set of drug test within a limited time, we hope current results is also informative to the community.*

b. How many replicates were measured for the dose response curves in Figure 5 a,b and c?

Response 4b: *We conducted three replicates for all drug tests.*

c. The dose response curves would be easier to interpret if the data was plotted in uM but on a log scale (rather than log transformed values).

Response 4c: *As reviewer mentioned, the dose response curves were plotted in uM.*

d. Why are there multiple values for MET copy number for each organoid in panel 5d?

Response 4d: *The gene copy number was estimated by oncopanel targeted capture sequencing, in which included multiple probes for capturing MET gene. Capture efficiency of each probe is different. Therefore, copy number value displayed as multiple values representing multiple probes covering MET gene.*

5. The authors indicate that LC organoids could be used for personalized care, so called patient ‘avatars’. This may be true in the future but this has not been demonstrated in this manuscript and a more nuanced presentation of their work would be more appropriate.

Response 5: *We modified the words “patient avatar” as “screening tool”.*

a. The Title of the manuscript should be altered because the authors do not demonstrate that these are personalised models that could inform the care of an individual patient.

Response 5a: *We respect the reviewer’s opinion. However, patient derived xenograft models and organoids are already widely accepted as a personalized cancer model^{7,8}.*

b. What is the timeline to generate organoids suitable for drug testing? How would this fit with the clinical path for a patient?

Response 5b: *To grow organoids suitable for drug testing, it usually needed to culture for 3 weeks. After this research, we did organoid culture from a small biopsy of metastatic adenocarcinoma of lung origin. In this case, organoids reached a suitable size for drug testing at 3 weeks in culture. To evaluate drug responses, it takes about 1 week. Therefore, we predict a total time from biopsy date to generate organoids and evaluate drug responses to the organoids to be about 4 weeks. In general, clinical setting, anticancer drug selection based on biomarker screening such as genomic sequencing and immunohistochemistry takes 4 weeks after biopsy date.*

c. How do the authors believe a personalised LC organoid approach would be used in the clinic? Many LC patients are cured surgically and therefore do the authors imagine this would be implemented in the setting of recurrent or metastatic disease? If so, are the authors able to derive LC organoids from fine needle biopsy samples?

Response 5c: *After submission of this manuscript, we generated organoids from a small biopsy of metastatic carcinoma. In this case organoids reached suitable size for drug testing at 3 weeks. As mentioned above therefore, in cases of metastatic disease or late stage inoperable disease, the organoid approach is comparable to the general clinical setting based on biomarker screening. In operable cases, as we mentioned in main text, approximately 26 % of stage I and II non-small-cell lung cancers recur at local regions or distant sites after curative surgery. The recurrence rate of stage III carcinoma is much higher. Therefore, organoid biobanking from resected lung cancer will be benefit for patients with high risk of recurrence.*

Reviewer #3 (Expertise: Lung cancer, Remarks to the Author):

In their manuscript, Kim and colleagues describe an approach for derivation of patient-specific lung cancer organoids that appears broadly suitable for culturing epithelial cells from 5 major cancer sub-types. They demonstrate the potential of these organoids for testing targeted molecular therapies and show that in the majority of cases the mutational spectrum of the cells is maintained *ex vivo*. They also show an enhanced capacity and rapidity of these cultured organoids to initiate tumors in subcutaneous xenotransplantation assays compared with conventional (non-3D) approaches. The work is of high quality and the findings are credible, but there is no new biology revealed, so the manuscript seems better suited as a technical report for lung cancer researchers rather than a research article.

Major criticism:

Because sequencing-based approaches can already identify the appropriate targeted therapy for the oncogenic mutations in any given patient, culture-based assays are currently only potentially useful for research applications. In this regard, there is no evidence provided that the ability to grow cancers as organoids provides any additional value over routine 2D culture, beyond establishing PDX models.

Response: *We respect the reviewer's personal opinion. However, we believe that this is not a specific issue related to our manuscript. It would be a more general issue towards all cancer organoid work that is rapidly transforming the cancer field. There are many research and review articles regarding the advantages of organoids comparing to 2D cells and PDX. In this manuscript, the most relevant finding is presented in Fig5. Exactly the same EGFR mutation was found in two cancer organoid lines. However, our drug tests revealed that the two cancer organoids have significantly different responses to Erlotinib and Crizotinib. NGS analysis may predict patient eligibility for targeted therapies with relatively high precision. However, within a class of targeted therapies, multiple drugs may be available as single or combinatorial agents, and may elicit differential responses from each individual cancer. For this purpose, it is extremely beneficial to grow patient-derived cancer materials with high success rate. Previous 2D cell lines and PDX models suffered from low establishment rates. PDXs were relatively better in the establishment rate, but are time-consuming and costly due to animal husbandry. All these issues have been well pointed out in this recent commentary (https://www.nature.com/articles/d41586-018-05890-8?fbclid=IwAR0_zmCu1hpSe8-ayt14lydYJwPH-dCNPγM-Mmdy6lRmGcUFsnPzFcVSMWQ), In our particular case, 2D cell lines established from patient samples had a lifespan of only 5 passages, whereas our organoid models could be passaged long term, cryopreserved, and thawed without functional consequence. We believe this demonstrates the advantage of the biobank in preclinical modelling. .*

Minor criticism:

Mutations in one of the organoids (LC-30) are reported to have zero concordance with mutations in the parental cancer. What do the authors make of this result? Does it suggest that the tumour cells did not grow and that the organoid was derived entirely of non-malignant lung epithelial cells admixed with the tumour? In any case, this result is concerning that the application of this organoid assay for patient-specific drug testing (or other applications) may be risky without first sequencing every organoid to confirm it is comprised of malignant cells.

Response: *Lung cancer is sometimes extremely heterogeneous, and sequencing also sometimes discordant between areas even in a single tumour. Our LCOs were generated from a very small portion of the tumour, which sometimes may not totally represent its original tumour. Well known cancer driver genes such as TP53, RB and EGFR usually show less intratumoural heterogeneity. LC-30 tissue and LCO-30 detected one gene (BRCA1) and three genes (PIK3CA, ERBB4 and FLT1) alterations, respectively. Unfortunately, all 4 genes are not well known driver gene of lung cancer. It is one explanation of the discordance. We cannot completely exclude the possibility of non-biological cause such as technical error. In our culture condition, normal cells cannot form organoids because of depletion of critical growth factors, as demonstrated in figure 3. We believe that we have many good examples of concordance between cancer organoids and the parental cancer. The reviewer is heavily concerned with one example out of 12 cases in targeted sequencing, where other cases are pretty promising. Moreover, we investigated whole exon sequencing (WES) in 7 cases of cancer organoids and their parental cancer tissues. They also showed the high concordance between cancer organoids and the parental cancer. We agree that one has to be always careful in using an “avatar” model so that the model can nicely recapitulate the original cancer tissue. Nevertheless, this kind of risk can be minimized by accompanying DNA analysis. Moreover, a similar bottleneck effect is shared in all cancer-derived models including cell lines and PDX models. We believe that cancer organoid models are relatively better in this aspect as we see well-preserved molecular heterogeneity between cancer organoids and the parental cancer (**Figure 3 in the revised manuscript**). For example, mutations with low representation in the cancer are also low in the organoids. Please note that due to nature of our organoid model that cancer cells grow without stroma, commonly present mutations reached to $MAF=1$ in the organoid part (see our explanation to Rev1-Q6).*

Reference

1. J. Z, *et al.* Sensitivity of neoplastic cells to senescence unveiled under standard cell culture conditions. *Anticancer Res* **35**, 2759–2768 (2015).
2. E. S-F, *et al.* The Failure in the Stabilization of Glioblastoma-Derived Cell Lines: Spontaneous In Vitro Senescence as the Main Culprit. *PloS one* **9**, e87136 (2014).
3. Houston KA, Mitchell KA, King J, White A, Ryan BM. Histologic Lung Cancer Incidence Rates and Trends Vary by Race/Ethnicity and Residential County. *Journal of thoracic oncology : official publication of the International Association for the Study of Lung Cancer* **13**, 497-509 (2018).
4. YC S, YC H, CY C. Role of TTF-1, CK20, and CK7 immunohistochemistry for diagnosis of primary and secondary lung adenocarcinoma. *Kaohsiung J Med Sci* **22**, 14-19 (2006).
5. Hiemstra PS, Bourdin A. Club cells, CC10 and self-control at the epithelial surface. *Eur Respir J* **44**, 831-832 (2014).
6. LL S, L L, WH Z, WD H. Establishment and identification of human primary lung cancer cell culture in vitro. *Int J Clin Exp Pathol* **8**, 6540-6546 (2015).
7. Grandori C, Kemp CJ. Personalized Cancer Models for Target Discovery and Precision Medicine. *Trends in cancer* **4**, 634-642 (2018).
8. Pauli C, *et al.* Personalized In Vitro and In Vivo Cancer Models to Guide Precision Medicine. *Cancer discovery* **7**, 462-477 (2017).

Reviewers' comments:

Reviewer #1 (Remarks to the Author):

In their study Kim et al report the generation and characterization of 80 patient-derived lung cancer organoids and 5 normal bronchial organoids derived from normal tumour-adjacent tissue. The authors used these organoids to perform proof-of-principle drug screening experiments in which they show that lung cancer organoids can potentially be used to predict patient response.

During the revisions of the manuscript, a paper appeared in EMBO journal by Sachs et al. This paper is briefly mentioned in this manuscript, yet provides very similar data. A more detailed comparison of the two studies appears appropriate. Some other issues remain

Major points

- In response 1, the authors state that they include more comprehensive images of the organoids growing in 3D. The images provided are still 'zoom-in' figures of a single organoid and do not show multiple organoids. Figure 4a, supplementary figure 1 and supplementary figure 7 would greatly improve if low-magnification images (each with at least a dozen organoids) is shown.
- In response 5d, the authors claim to show CD56 and synaptophysin positivity of their organoids and tissue. Synaptophysin staining is not shown and CD56 staining appears negative in their organoids. It is therefore not convincingly shown that these organoids represent the claimed large cell neuroendocrine tumour subtype. Please remove this entirely or replace by better data

Minor points

- In line 240, the authors reference figure 4b while this does not show cell type specific labelling. The reference should be extended to 4b-c
- In response 3, the authors bring forth valuable data about the improvement of organoids over 2D cell lines. This should be included in the main text as well.

Reviewer #2 (Remarks to the Author):

The authors have substantively addressed my comments and overall the manuscript is significantly improved and suitable for publication. I remain of the opinion, as indicated in my original comments, that the title should be altered to remove the work personalized. From my original review:

"a. The Title of the manuscript should be altered because the authors do not demonstrate that these are personalised models that could inform the care of an individual patient.

Response 5a: We respect the reviewer's opinion. However, patient derived xenograft models and organoids are already widely accepted as a personalized cancer model^{7, 8}.

Despite the response from the authors, they have not yet demonstrated that their LC organoids could inform personalised/individualised patient care. Others have attempted this with some success (Georgios Vlachogiannis et al. Science 2018), whereas others have found this challenging to implement (Pauli et al Cancer Discovery. 2017). I agree that this is an worthwhile aspiration and their work contribute to this goal, but this has not been demonstrated here.

Reviewer #3 (Remarks to the Author):

The revised manuscript by Kim and colleagues is overall much improved. They have provided

several pieces of data demonstrating how their 3D culture system is superior than existing 2D methods, including a higher likelihood of establishing the culture, much longer passaging capacity with maintenance of genetic stability, and increased efficiency of xenotransplantation. The drug experiments correlating sensitivity with observed mutations are also well-fleshed out and the outcomes largely match the expected results. My concerns from the initial submission have been adequately addressed.

Response to referees letter

We greatly appreciate the reviewers' thoughtful and insightful suggestions. We are sure that additional revisions as suggested by reviewers have significantly improved quality of our manuscript. All expressed concerns have been addressed with necessary changes made to the manuscript. Our specific responses are following:

Specific responses to reviewers

Reviewer #1 (Expertise: Cancer organoid biobank, Remarks to the Author):

In their study Kim et al report the generation and characterization of 80 patient-derived lung cancer organoids and 5 normal bronchial organoids derived from normal tumour-adjacent tissue. The authors used these organoids to perform proof-of-principle drug screening experiments in which they show that lung cancer organoids can potentially be used to predict patient response.

During the revisions of the manuscript, a paper appeared in EMBO journal by Sachs et al. This paper is briefly mentioned in this manuscript, yet provides very similar data. A more detailed comparison of the two studies appears appropriate. Some other issues remain.

Response: *As reviewer mentioned, we added additional descriptions covering a detailed comparison of EMBO journal by Sachs et al. and our work. A certain amount of normal cells are usually included in a cancer tissue harvested for culture. As growth of normal cells are faster than cancer cells in a high nutritious optimal medium condition, they hamper the growth of cancer organoids. For this reason, previous papers mentioned that cancer organoids need to be cultured in a culture medium lacking some growth factors(a sub-optimal medium)^{1, 2}. However, in their paper, they did not show the sub-optimal medium condition to prevent outgrowth of normal cells, and mentioned that they could not selectively expand lung tumouroids by removing a single medium components. Instead, they manually separated normal airway organoids mixed in cancer organoids based on organoid morphology. In case of p53 mutated cancer, they used medium containing Nutulin-3a to remove normal cells having wild type p53. In our manuscript, we described a well-designed sub-optimal medium for high-efficiency culture of lung cancer organoids without being bothered with normal cell contamination and an optimal culture medium for normal bronchial organoids. And we showed that normal bronchial organoids could not grow using our sub-*

*optimal medium in **Figure 4a of the revised manuscript**. Manual separation is time consuming tedious work. And to use Nutulin-3a method, tumour genomic sequencing prior to organoid culture is required.*

Major points:

Comment 1: In response 1, the authors state that they include more comprehensive images of the organoids growing in 3D. The images provided are still 'zoom-in' figures of a single organoid and do not show multiple organoids. Figure 4a, supplementary figure 1 and supplementary figure 7 would greatly improve if low-magnification images (each with at least a dozen organoids) is shown.

Response 1: *As reviewer mentioned, we changed images showing multiple organoids in **Figure 4a, supplementary figure 1 and supplementary figure 7 of the revised manuscript**.*

Comment 2: In response 5d, the authors claim to show CD56 and synaptophysin positivity of their organoids and tissue. Synaptophysin staining is not shown and CD56 staining appears negative in their organoids. It is therefore not convincingly shown that these organoids represent the claimed large cell neuroendocrine tumour subtype. Please remove this entirely or replace by better data.

Response 2: *As reviewer mentioned, we removed this data. Although cancer organoids mimic histologic features of large cell neuroendocrine carcinoma tissue, we fail to recapitulate expression of neuroendocrine marker CD56. We could not clearly explain the reason in this paper. Another culture condition adding some growth factor such as Notch³ may improve to induce neuroendocrine differentiation.*

Minor points

Comment 3:

In line 240, the authors reference figure 4b while this does not show cell type specific labelling. The reference should be extended to 4b-c.

Response 3: *As reviewer mentioned, we added the reference **in the revised manuscript**.*

Comment 4:

In response 3, the authors bring forth valuable data about the improvement of organoids over 2D cell lines. This should be included in the main text as well.

Response 4: *As reviewer mentioned, we described it **on p.13 line 14-22 in the discussion section of the revised manuscript**.*

Reviewer #2 (Expertise: Cancer genomics, Remarks to the Author):

The authors have substantively addressed my comments and overall the manuscript is significantly improved and suitable for publication. I remain of the opinion, as indicated in my original comments, that the title should be altered to remove the work personalized. From my original review:

Comment 1:

The authors have substantively addressed my comments and overall the manuscript is significantly improved and suitable for publication. I remain of the opinion, as indicated in my original comments, that the title should be altered to remove the work personalized. From my original review:

5a. The Title of the manuscript should be altered because the authors do not demonstrate that these are personalised models that could inform the care of an individual patient.

Response 5a: We respect the reviewer's opinion. However, patient derived xenograft models and organoids are already widely accepted as a personalized cancer model^{4,5}.

Despite the response from the authors, they have not yet demonstrated that their LC organoids could inform personalised/individualised patient care. Others have attempted this with some success (Georgios Vlachogiannis et al. Science 2018), whereas others have found this challenging to implement (Pauli et al Cancer Discovery. 2017). I agree that this is a worthwhile aspiration and their work contribute to this goal, but this has not been demonstrated here.

Response 1: We agree with the reviewer's opinion. We changed the title of the revised manuscript to "Patient-Derived Lung Cancer Organoids as In Vitro Cancer Models for Therapeutic Screening"

Reviewer #3 (Expertise: Lung cancer, Remarks to the Author):

The revised manuscript by Kim and colleagues is overall much improved. They have provided several pieces of data demonstrating how their 3D culture system is superior than existing 2D methods, including a higher likelihood of establishing the culture, much longer passaging capacity with maintenance of genetic stability, and increased efficiency of xenotransplantation. The drug experiments correlating sensitivity with observed mutations are also well-fleshed out and the outcomes largely match the expected results. My concerns from the initial submission have been adequately addressed.

Reference

1. Drost J, Clevers H. Organoids in cancer research. *Nature reviews Cancer* **18**, 407-418 (2018).
2. Weeber F, Ooft SN, Dijkstra KK, Voest EE. Tumor Organoids as a Pre-clinical Cancer Model for Drug Discovery. *Cell chemical biology* **24**, 1092-1100 (2017).
3. Nakakura EK, *et al.* Regulation of neuroendocrine differentiation in gastrointestinal carcinoid tumor cells by notch signaling. *The Journal of clinical endocrinology and metabolism* **90**, 4350-4356 (2005).
4. Grandori C, Kemp CJ. Personalized Cancer Models for Target Discovery and Precision Medicine. *Trends in cancer* **4**, 634-642 (2018).
5. Pauli C, *et al.* Personalized In Vitro and In Vivo Cancer Models to Guide Precision Medicine. *Cancer discovery* **7**, 462-477 (2017).

REVIEWERS' COMMENTS:

Reviewer #1 (Remarks to the Author):

The authors have greatly improved their manuscript by addressing the reviewer's comments. The manuscript is suitable for publication and will have a major contribution to the field of lung organoids and lung cancer. However, two minor points remain unaddressed.

Minor point 1:

The authors' response towards the comparison with Sachs et al. are valid. However, the introduction of the manuscript is not referencing the airway organoid paper while it is appropriate. If the authors address this in for example line 72 and line 82 the impact of this manuscript will be better put in perspective of current literature.

Minor point 2:

The authors claim to show more comprehensive images of their culture wells. While there are more organoids visible in figure 4 and figure S2, there is still not a clear view on the general condition of most of the organoids in the well. The authors, however, do show comprehensive images in supplementary figure 7. This zoom is needed in earlier mentioned figure to increase the claim of a viable culture.

Response to referee's letter

We greatly appreciate the reviewers' thoughtful and insightful suggestions. We are sure that additional revisions as suggested by reviewers have significantly improved quality of our manuscript. All expressed concerns have been addressed with necessary changes made to the manuscript. Our specific responses are following:

REVIEWERS' COMMENTS:

Reviewer #1 (Remarks to the Author):

The authors have greatly improved their manuscript by addressing the reviewer's comments. The manuscript is suitable for publication and will have a major contribution to the field of lung organoids and lung cancer. However, two minor points remain unaddressed.

Response: *We thanks to reviewer's encouraging comments.*

Minor point 1:

The authors' response towards the comparison with Sachs et al. are valid. However, the introduction of the manuscript is not referencing the airway organoid paper while it is appropriate. If the authors address this in for example line 72 and line 82 the impact of this manuscript will be better put in perspective of current literature.

Response: *As reviewer mentioned, we referenced the previous paper of Sachs et al. describing airway organoids and lung cancer organoids in the introduction on p.4 lines 8-12 in the introduction section of the revised manuscript.*

Minor point 2:

The authors claim to show more comprehensive images of their culture wells. While there are more organoids visible in figure 4 and figure S2, there is still not a clear view on the general condition of most of the organoids in the well. The authors, however, do show comprehensive images in supplementary figure 7. This zoom is needed in earlier mentioned figure to increase the claim of a viable culture.

Response 2: *As reviewer mentioned, we changed the images of organoids in **Supplementary Figure 2 in the revised manuscript**. However, regarding figure 4a, we could not change the pictures by more comprehensive images of the organoids in the well. When we cultivate the normal bronchial organoids for*

this experiment, we mainly focused on budding structures in normal airway organoids and unfortunately we did not take photos for comprehensive images. We only could change one picture of Day 14, passage 1. We sincerely hope this minor change and explanation will be acceptable for the reviewer.